# Evaluation of Best-of-N Sampling Strategies for Language Model Alignment

**Yuki Ichihara**  *ichihara.yuki.iu1@is.naist.jp*
*Nara Institute of Science and Technology*

**Yuu Jinnai**  *jinnai_yu@cyberagent.co.jp*
**Tetsuro Morimura**  *morimura_tetsuro@cyberagent.co.jp*
**Kenshi Abe**  *abe_kenshi@cyberagent.co.jp*
**Kaito Ariu**  *kaito_ariu@cyberagent.co.jp*
**Mitsuki Sakamoto**  *sakamoto_mitsuki@cyberagent.co.jp*
*CyberAgent*

**Eiji Uchibe**  *uchibe@atr.jp*
*Advanced Telecommunications Research Institute International*

**Reviewed on OpenReview:** *https://openreview.net/forum?id=H4S4ETc8c9*

## Abstract

Best-of-N (BoN) sampling with a reward model has been shown to be an effective strategy for aligning Large Language Models (LLMs) with human preferences at the time of decoding. BoN sampling is susceptible to a problem known as *reward hacking*. Since the reward model is an imperfect proxy for the true objective, an excessive focus on optimizing its value can lead to a compromise of its performance on the true objective. Previous work proposes Regularized BoN sampling (RBoN), a BoN sampling with regularization to the objective, and shows that it outperforms BoN sampling so that it mitigates reward hacking and empirically (Jinnai et al., 2024). However, Jinnai et al. (2024) introduce RBoN based on a heuristic and they lack the analysis of *why* such regularization strategy improves the performance of BoN sampling. The aim of this study is to analyze the effect of BoN sampling on regularization strategies. Using the regularization strategies corresponds to robust optimization, which maximizes the worst case over a set of possible perturbations in the proxy reward. Although the theoretical guarantees are not directly applicable to RBoN, RBoN corresponds to a practical implementation. This paper proposes an extension of the RBoN framework, called Stochastic RBoN sampling (SRBoN), which is a theoretically guaranteed approach to worst-case RBoN in proxy reward. We then perform an empirical evaluation using the AlpacaFarm and Anthropic's hh-rlhf datasets to evaluate which factors of the regularization strategies contribute to the improvement of the true proxy reward. In addition, we also propose another simple RBoN method, the Sentence Length Regularized BoN, which has a better performance in the experiment as compared to the previous methods.

## 1 Introduction

Large Language Models (LLMs) have demonstrated remarkable capabilities in many NLP tasks related to natural language understanding and text generation (Stiennon et al., 2020; Ouyang et al., 2022; Touvron et al., 2023; Dubey et al., 2024; OpenAI et al., 2024). Despite the strengths, LLMs are not always adept at interpreting a wide range of instructions and can produce undesirable outputs, such as biased, hallucinated, or toxic responses (Bai et al., 2022; Lin et al., 2022; Touvron et al., 2023; Casper et al., 2023; Guan et al., 2024). This problem underscores the challenge of language model alignment; ensuring LLMs' behaviors align with human objectives and safety considerations (Ziegler et al., 2020; Stiennon et al., 2020; Ouyang et al.,

2022). There is now a rich set of approaches to address this problem (Stiennon et al., 2020; Ouyang et al., 2022; Rafailov et al., 2023). These papers have shown that training language models with human feedback can improve their performance. However, training language models is a computationally intensive task. In other words, improved performance comes at the cost of increased computational resources.

This paper focuses on the Best-of-N (BoN) sampling strategy, a method that involves generating $N$ outputs from a model and selecting the most preferred output among the $N$ samples. Despite its simplicity and the fact that it does not require an additional training phase, BoN sampling has been shown to be surprisingly effective in practice (Stiennon et al., 2020; Nakano et al., 2022). However, the BoN strategy does not scale with the number of samples $N$ due to the *reward hacking problem* (Amodei et al., 2016; Ziegler et al., 2020; Stiennon et al., 2020; Skalse et al., 2022; Gao et al., 2023). Reward hacking is a behavior that satisfies the given objective without achieving the intended result. It is caused by the misspecification of the preference model used by the BoN to select the most preferred output (Pan et al., 2022; Lambert & Calandra, 2024).

Previous work to mitigate the reward hacking problem has proposed Regularized Best-of-N sampling (RBoN), BoN strategy with the addition of a regularization term to the objective (Jinnai et al., 2024). This paper shows that the RBoN strategy is effective compared to BoN sampling in various experiments. However, *why* such a regularization strategy is effective against reward uncertainty is unclear in the previous work.

In this paper, we propose Stochastic RBoN sampling (SRBoN), which adds a regularization term similar to RBoN. We then draw a connection between the Reinforcement Learning (RL) problems (Sutton & Barto, 2018) and the BoN strategies: BoN sampling corresponds to solving the RL problem, and SRBoN sampling strategies correspond to solving the Regularized Reinforcement Learning (RRL) problem (Neu et al., 2017; Geist et al., 2019; Yang et al., 2019; Derman et al., 2021).

First, we exploit the knowledge of RRL. Some work has shown that regularization terms in probability distributions over outputs provide robustness to reward perturbations (Ortega & Lee, 2014; Husain et al., 2021; Derman et al., 2021; Eysenbach & Levine, 2022; Pan et al., 2022; Derman et al., 2021). SRBoN can also apply this analysis to RRL. Its insights provide an answer to why reward hacking can be mitigated: when a regularization term is added to the BoN sampling, it also becomes an adversarial perturbation to the reward.

We then evaluate the effectiveness of our approach against alternative decoder methods in a series of experiments, with the goal of determining the relative resilience of each method to potential exploitation by reward hacking. The results show that our proposed method outperforms many existing approaches in a variety of settings. In other words, a theoretically guaranteed, effective algorithm is proposed.

In addition, while RBoN consists of complex formula structures, we proposed a simpler RBoN, Sentence Length Regularized BoN that, despite its simple implementation, shows comparable or even better performance in experiments with the methods of previous studies.

## 2 Background

In this paper, we formalize the problem of decoding time alignment as Regularized Markov Decision Processes (MDPs) problem (Neu et al., 2017; Geist et al., 2019; Yang et al., 2019; Derman et al., 2021). For brevity, we refer to reinforcement learning within Regularized MDPs as Regularized Reinforcement Learning (RRL) throughout this paper. In Section 2.1, we describe the Reinforcement Learning (RL) problem and the RRL problem. Then, we describe two sampling algorithms used for decoding time alignment, Best-of-N (BoN) sampling in Section 2.2, and the Regularized Best-of-N sampling (RBoN) in Section 2.3.

### 2.1 Adversarial Interpretation in Regularized Reinforcement Learning

We consider the problem of selecting an output $y$ from a set of outputs $\mathcal{Y}_{\mathbf{ref}} \subseteq \mathcal{Y}$ (e.g., response text from the system) given an input $x \in \mathcal{X}$ (e.g., input prompt by a user), where the objective is to select the best output according to a reward function $R \colon \mathcal{X} \times \mathcal{Y}_{\mathbf{ref}} \to \mathbb{R}$. Let $\Delta(\mathcal{Y})$ denote the set of probability distributions over a set $\mathcal{Y}$. We define the goal of the Reinforcement Learning (RL) problem as finding the best policy

$\pi : \mathcal{X} \to \Delta(\mathcal{Y}_{\mathbf{ref}})$ that maximizes the expected reward for $x$:

$$
\begin{aligned}
\arg\max_{\pi \in \Pi} f_{\mathrm{RL}}(\pi) &= \arg\max_{\pi \in \Pi} \sum_{y \in \mathcal{Y}_{\mathbf{ref}}} \pi(y \mid x) R(x, y) \\
&= \arg\max_{\pi \in \Pi} \mathbb{E}_{y \sim \pi(\cdot \mid x)}[R(x, y)]
\end{aligned}
\tag{1}
$$

where $\Pi$ is a set of all possible policies. Note that there exists a deterministic policy that maximizes $f_{\mathrm{RL}}$ (Sutton & Barto, 2018) and this formulation can be seen as a contextual bandit problem. Specifically, the policy observes a context $x$, chooses an output $y$ based on that context, and receives a reward $R(x, y)$. Importantly, we do not use sequential decision operations or consider state transitions. Each decision is made independently based on the current context $x$.

The underlying assumption of the RL problem is that the reward model $R$ is correctly defined and observable. That is, we consider the solution that maximizes the expected reward to be the optimal solution. However, real-world applications often suffer from the *reward misspecification problem* – the reward model observable to the agent is only a proxy for the true underlying reward of the problem (Ortega & Lee, 2014; Husain et al., 2021; Derman et al., 2021; Eysenbach & Levine, 2022; Pan et al., 2022; Derman et al., 2021). Prior work has investigated strategies to optimize under the uncertainty in the observed reward. In contrast to the RL problem, Regularized Reinforcement Learning (RRL) incorporates regularization terms to achieve a solution that is robust to the reward misspecification (Neu et al., 2017; Geist et al., 2019; Yang et al., 2019; Derman et al., 2021). The objective of the RRL problem is to find the best policy $\pi : \mathcal{X} \to \Delta(\mathcal{Y}_{\mathbf{ref}})$ that maximizes the reward with an additional regularization function $\Omega(\pi) : \Delta(\mathcal{Y}_{\mathbf{ref}}) \to \mathbb{R} \cup \{+\infty\}$. Let $f_{\mathrm{RRL}}^{\Omega}(\pi) := \mathbb{E}_{y \sim \pi(\cdot \mid x)}[R(x, y)] - \Omega(\pi)$ be the objective function of RRL problem with $\Omega$. Then, we define the following as the optimal solution to the RRL problem:

$$
\arg\max_{\pi \in \Pi} f_{\mathrm{RRL}}^{\Omega}(\pi) = \arg\max_{\pi \in \Pi} \mathbb{E}_{y \sim \pi(\cdot \mid x)}[R(x, y)] - \Omega(\pi).
\tag{2}
$$

Note that, unlike the RL problem, there may not exist an optimal policy that is deterministic for the RRL problem (Geist et al., 2019).

Brekelmans et al. (2022) uses Legendre–Fenchel transformation (Touchette, 2005) to show that the RRL problem can be viewed as a variant of the RL problem with an adversarial agent adding perturbations to the reward $\Delta R : \mathcal{X} \times \mathcal{Y}_{\mathbf{ref}} \to \mathbb{R}$ if the regularization term $\Omega$ is convex and lower semi-continuous function (Boyd & Vandenberghe, 2004):

$$
\arg\max_{\pi \in \Pi} f_{\mathrm{RRL}}^{\Omega}(\pi) = \arg\max_{\pi \in \Pi} \min_{\Delta R \in \mathcal{R}_{\Delta}} \mathbb{E}_{y \sim \pi(\cdot \mid x)}[R(x, y) - \Delta R(x, y)] + \Omega^{*}(\Delta R),
\tag{3}
$$

where $\Omega^{*} \in \mathbb{R}^{\mathcal{X} \times \mathcal{Y}_{\mathbf{ref}}}$ is the conjugate function of $\Omega$ (Boyd & Vandenberghe, 2004), and $\mathcal{R}_{\Delta} := \{\Delta R \in \mathbb{R}^{\mathcal{X} \times \mathcal{Y}_{\mathbf{ref}}} \mid F_{\Omega}(\Delta R) \leq 0\}$, where $F_{\Omega}$ is a function or operator dependent to $\Omega$ that imposes a constraint or condition on the values of $\Delta R$.

Eq. (3) shows that the problem of maximizing $f_{\mathrm{RRL}}$ can be reformulated as the max-min problem and the regularizer $\Omega$ is effectively an adversarial reward perturbation that forces us to optimize the worst case performance (Derman et al., 2021).

## 2.2 Best-of-N (BoN) Sampling

BoN sampling has emerged as an effective method for preference optimization in LLMs (Stiennon et al., 2020; Nakano et al., 2022). BoN sampling has several advantages over preference learning methods. First, it is straightforward and does not require additional training in the language model. Although learning-based alignment methods require retraining the LLMs whenever human preferences are updated, BoN sampling can be applied immediately, requiring only an update of the reward model. This is particularly advantageous since training LLMs is the most resource-intensive process. Second, BoN sampling is an effective strategy in its own right, with numerous studies demonstrating that it can outperform learning-based adaptation methods (Gao et al., 2023). Recent literature has expanded our understanding of BoN sampling. In particular, Beirami

et al. (2024) conducted an analysis comparing the policies selected by BoN sampling with the base policies used for sample generation. In addition, Gui et al. (2024) showed that BoN sampling achieves an optimal balance between win rate and KL divergence when aligning large language models to human preferences.

BoN sampling has similarities to the objective function used in RL (e.g., the response with the highest reward score, determined by a proxy reward model $R(x, y)$, is selected). The objective function of BoN is given by:

$$y_{\text{BoN}}(x) := \arg\max_{y \in \mathcal{Y}_{\mathbf{ref}}} R(x, y).$$

This reward model is used as a measure of the quality of the text when making an assignment. The reward model we consider in this paper is open access as described in Appendix I. We also mention that the objective function of BoN sampling is equal to the objective function of the (unregularized) RL problem (Eq. (1)):

$$
\begin{aligned}
y_{\text{BoN}}(x) : &= \arg\max_{y \in \mathcal{Y}_{\mathbf{ref}}} R(x, y) \\
&= \arg\max_{y \in \mathcal{Y}_{\mathbf{ref}}} \max_{\pi_y \in \Pi_{det}} \mathbb{E}_{y \sim \pi_y}[R(x, y)] \\
&= \arg\max_{y \in \mathcal{Y}_{\mathbf{ref}}} \max_{\pi_y} f_{\text{RL}}(\pi_y).
\end{aligned}
\tag{4}
$$

where $\Pi_{det}$ is a set of deterministic policies and $\pi_y$ is a deterministic policy that selects $y$ given $x$ with a probability of 1.

## 2.3 Regularized Best-of-N Sampling (RBoN)

Although BoN sampling is shown to be effective, it is prone to the reward hacking problem (Amodei et al., 2016; Ziegler et al., 2020; Stiennon et al., 2020; Skalse et al., 2022; Gao et al., 2023). The reward hacking problem is a phenomenon where the decision to optimize the proxy reward is made without considering its potential misspecification, resulting in worse performance on the actual reward objective. Dubois et al. (2023) showed that with 25% label noise, which is the amount of disagreement observed in real-world preference annotations (Stiennon et al., 2020; Ouyang et al., 2022), BoN sampling degrades performance with $N$ greater than 16 (Figures 12 and 13 in Dubois et al. 2023).

Regularized Best-of-N sampling (RBoN) is proposed to mitigate the reward hacking problem for BoN sampling (Jinnai et al., 2024). Jinnai et al. (2024) presented two variants of RBoN: using the KL divergence as a regularizer (RBoN$_{\text{KL}}$; Section 2.3.1) and using the Wasserstein Distance as a regularizer (RBoN$_{\text{WD}}$; Section 2.3.2). In the following, we describe the two variants of RBoN and draw its connection to the objective function of RRL (Eq. (2)).

### 2.3.1 KL divergence Regularized BoN Sampling (RBoN$_{\text{KL}}$)

The objective function of RBoN$_{\text{KL}}$ (KL divergence Regularized BoN Sampling) is given by:

$$
\begin{aligned}
y_{\text{KLBoN}}(x) &= \arg\max_{y \in \mathcal{Y}_{\mathbf{ref}}} R(x, y) - \beta D_{\text{KL}}\left[\pi_y(\cdot \mid x) \| \pi_{\mathbf{ref}}(\cdot \mid x)\right], \\
&= \arg\max_{y \in \mathcal{Y}_{\mathbf{ref}}} \max_{\pi_y \in \Pi_{det}} f_{\text{RRL}}^{\text{KL}}(\pi_y).
\end{aligned}
$$

where $\beta$ is a regularization parameter, reference policy $\pi_{\mathbf{ref}}$: $\mathcal{X} \to \Delta(\mathcal{Y}_{\mathbf{ref}})$, and $D_{\text{KL}}$ denotes the KL divergence. The reference policy here takes an input x and returns a meaningful output y. The regularization described in this paper is like a penalty to stay away from the reference policy and the reference policy is the language model.

By incorporating the KL divergence as a regularization term in the objective function, RBoN$_{\text{KL}}$ encourages the learned policy to be close to the reference policy $\pi_{\mathbf{ref}}$. A higher value of $\beta$ emphasizes the regularization term, encouraging the learned policy to be closer to the reference policy, while a lower value of $\beta$ prioritizes maximizing the reward function.

### 2.3.2 Wasserstein Distance Regularized BoN Sampling ($\mathrm{RBoN_{WD}}$)

The objective function of $\mathrm{RBoN_{WD}}$ (Wasserstein Distance Regularized BoN Sampling) is defined as follows:

$$y_{\mathrm{WDBoN}}(x) = \underset{y \in \mathcal{Y}_{\mathbf{ref}}}{\arg\max} \; R(x, y) - \beta \mathbf{WD}\left[\pi_y(\cdot \mid x) \| \pi_{\mathbf{ref}}(\cdot \mid x)\right],$$

$$= \underset{y \in \mathcal{Y}_{\mathbf{ref}}}{\arg\max} \; \underset{\pi_y \in \Pi_{det}}{\max} \; f_{\mathrm{RRL}}^{\mathrm{WD}}(\pi_y).$$

where $\mathbf{WD}$ denotes 1-Wasserstein Distance.

The $\mathbf{WD}$ (Wang, 2012) is defined as:

$$\mathbf{WD}[\nu \| \mu] = \inf_{\gamma \in \Gamma(\nu, \mu)} \sum_{(i,j) \in N \times N} \gamma_{ij}\, C_{ij}, \tag{5}$$

where $N$: the total number of samples, consisting of the set $\{y_1, y_2, \ldots, y_N\}$, $\nu, \mu \in \Delta(N)$: probability measure on the aforementioned sets ($\nu_i, \mu_i$ refer to the probability value $\nu(y_i), \mu(y_i)$), $C$: $N \times N \to \mathbb{R}$ a cost function measuring the distance between two outputs (e.g. $C_{ij}$ refers to the amount to be transported from place $y_i$ to palace $y_j$), and $\Gamma(\nu, \mu)$ denotes the set of all joint distributions $\gamma$ whose marginals are $\nu$ and $\mu$. The constraints on $\gamma$ are given by:

$$\sum_{j \in n} \gamma_{ij} = \nu_i, \quad \forall i \in n,$$

$$\sum_{i \in n} \gamma_{ij} = \mu_j, \quad \forall j \in n,$$

$$\gamma_{ij} \geq 0, \quad \forall i, j \in n.$$

The $\mathbf{WD}$, also known as the Earth Mover's Distance (EMD), is a metric used to quantify the dissimilarity between two probability distributions. Intuitively, it measures the minimum cost required to transform one distribution into the other. This cost is conceptualized as the amount of probability mass that must be moved multiplied by the distance that would be moved. The concept has been used in NLP to measure the dissimilarity of texts (Kusner et al., 2015; Zhao et al., 2019).

The exact computation of $\mathbf{WD}\left[\pi_y(\cdot \mid x) \| \pi_{\mathbf{ref}}(\cdot \mid x)\right]$ is intractable due to the enormous size of the output space. To address this computational challenge, prior work (Jinnai et al., 2024) has employed sample-based approximation techniques. Let $\hat{\pi}_{\mathbf{ref}}$ represent the empirical distribution computed using a set of samples $\mathcal{Y}_{\mathbf{ref}}$. This distribution is defined as: $\hat{\pi}_{\mathbf{ref}}(y \mid x) = \frac{1}{N} \sum_{y' \in \mathcal{Y}_{\mathbf{ref}}} \mathbb{I}(y = y')$ where $N$ is the total number of samples in $\mathcal{Y}_{\mathbf{ref}}$. The objective function can then be approximated as follows:

$$y_{\mathrm{WDBoN}}(x) = \underset{y \in \mathcal{Y}_{\mathbf{ref}}}{\arg\max} \; R(x, y) - \beta \mathbf{WD}\left[\pi_y(\cdot \mid x) \| \hat{\pi}_{\mathbf{ref}}(\cdot \mid x)\right].$$

For practical implementation aspects, the $\mathbf{WD}$ term for Jinnai et al. (2024) is computed as follows:

$$\mathbf{WD}\left[\pi_y(\cdot \mid x) \| \hat{\pi}_{\mathbf{ref}}(\cdot \mid x)\right] = \sum_{y' \in \mathcal{Y}_{\mathbf{ref}}} \frac{1}{N} C\left(y, y'\right), \tag{6}$$

The cosine distance is used as $C$ to measure the distance between the outputs (Reimers & Gurevych, 2019).

$$C\left(y, y'\right) = 1 - \cos\left(\mathrm{emb}(y), \mathrm{emb}\left(y'\right)\right), \tag{7}$$

where $\mathrm{emb}(y)$ and $\mathrm{emb}\left(y'\right)$ represent the embeddings of output $y$ and $y'$, respectively.

## 3 Stochastic RBoN (SRBoN)

We propose the stochastic version of RBoN, Stochastic $\mathrm{RBoN_{KL}}$ (Section 3.1) and Stochastic $\mathrm{RBoN_{WD}}$ (Section 3.2). These novel algorithms, while similar to the original RBoN (deterministic version), allow for the optimal policy $\pi$ to a probabilistic output distribution. By relaxing the deterministic constraint, we can apply theoretical tools that were previously inaccessible. Our approach focuses on the analysis of this stochastic version, aiming to provide theoretical results that shed light on the underlying mechanisms of RBoN's effectiveness.

### 3.1 Stochastic $\text{RBoN}_{\text{KL}}$ ($\text{SRBoN}_{\text{KL}}$)

First, consider a stochastic version of $\text{RBoN}_{\text{KL}}$. The policy of $\text{SRBoN}_{\text{KL}}$ is given by:

$$
\begin{aligned}
\pi_{\text{SRBoN}_{\text{KL}}}(x) &= \underset{\pi \in \Pi}{\arg\max}\, \mathbb{E}_{y \sim \pi(\cdot|x)}[R(x,y)] - \beta D_{\text{KL}}\left[\pi(\cdot \mid x) \| \pi_{\text{ref}}(\cdot \mid x)\right] \\
&= \underset{\pi \in \Pi}{\arg\max}\, f_{\text{RRL}}^{\text{KL}}(\pi).
\end{aligned}
\tag{8}
$$

We define $\text{SRBoN}_{\text{KL}}$ as a method to sample a response $y$ that follows the probability distribution of $\pi_{\text{SRBoN}_{\text{KL}}}$:

$$
y_{\text{SRBoN}_{\text{KL}}}(x) \sim \pi_{\text{SRBoN}_{\text{KL}}}(x).
\tag{9}
$$

In Section 4 we evaluate the performance of this stochastic text generation algorithm defined by Eq. (9).

#### 3.1.1 Theoretical Guarantee of $\text{SRBoN}_{\text{KL}}$

By relaxing the deterministic policy constraint of $\text{RBoN}_{\text{KL}}$, $\text{SRBoN}_{\text{KL}}$ follows the formulation of the RRL with adversarial perturbations studied by Brekelmans et al. (2022). As such, the computation of $\text{SRBoN}_{\text{KL}}$ can be transformed into a max-min problem using Legendre-Fenchel transformation (Touchette, 2005) as in Eq. (3). In this way, $\text{SRBoN}_{\text{KL}}$ has the following theoretical guarantee proven by Brekelmans et al. (2022):

**Theorem 3.1.** *(**Brekelmans et al. (2022), Proposition 1**) The problem of maximizing $f_{\text{RRL}}^{\text{KL}}(\pi)$ can be interpreted as a robust optimization problem with an adversarial perturbation $\Delta R$:*

$$
\underset{\pi \in \Pi}{\arg\max}\, f_{\text{RRL}}^{\text{KL}}(\pi) = \underset{\pi \in \Pi}{\arg\max}\, \underset{\Delta R \in \mathcal{R}_\Delta}{\min}\, \mathbb{E}_{y \sim \pi(\cdot|x)}[R(x,y) - \Delta R(x,y)],
\tag{10}
$$

*where the feasible set of reward perturbations $\mathcal{R}_\Delta$ available to the adversary is bounded:*

$$
\mathcal{R}_\Delta := \left\{ \Delta R \in \mathbb{R}^{\mathcal{X} \times \mathcal{Y}_{\text{ref}}} \mid \sum_{\mathcal{Y}_{\text{ref}}} \pi_{\text{ref}}(y \mid x) \exp(\beta^{-1} \Delta R(x,y)) \leq 1 \right\}
\tag{11}
$$

The theorem shows that $\text{SRBoN}_{\text{KL}}$ is an algorithm that optimizes the worst-case performance under the assumption that the error between the true reward and the given proxy reward model is guaranteed to be within $\mathcal{R}_\Delta$ (Eq. (11)).

Let $\mathcal{R}'$ be a set of possible reward models under the reward perturbations: $\mathcal{R}' := \{R - \Delta R \mid \Delta R \in \mathcal{R}_\Delta\}$. Let $f_{\text{RRL}}^{\text{KL}}(\pi; R)$ be the objective of the policy given a (proxy) reward model $R$. Then,

$$
\begin{aligned}
\underset{\pi \in \Pi}{\arg\max}\, f_{\text{RRL}}^{\text{KL}}(\pi; R) &= \underset{\pi \in \Pi}{\arg\max}\, \underset{\Delta R \in \mathcal{R}_\Delta}{\min}\, \mathbb{E}_{y \sim \pi(\cdot|x)}[R(x,y) - \Delta R(x,y)] \\
&= \underset{\pi \in \Pi}{\arg\max}\, \underset{R' \in \mathcal{R}'}{\min}\, \mathbb{E}_{y \sim \pi(\cdot|x)}[R'(x,y)] \\
&= \underset{\pi \in \Pi}{\arg\max}\, \underset{R' \in \mathcal{R}'}{\min}\, f_{\text{RRL}}^{\text{KL}}(\pi; R').
\end{aligned}
\tag{12}
$$

Thus, $\text{SRBoN}_{\text{KL}}$ is a robust optimization of the policy for a set of possible reward models in $\mathcal{R}'$. In other words, it assumes that the true payoff model is in $\mathcal{R}_\Delta$ and optimizes for the worst case.

The theorem is derived by translating the proposition proved by Brekelmans et al. (2022) for the generic RRL problems to the text generation scenario. The contribution of our work is to show the relation of their theoretical result to the RBoN sampling algorithm in LLMs alignment.

### 3.2 Stochastic $\text{RBoN}_{\text{WD}}$ ($\text{SRBoN}_{\text{WD}}$)

We now consider an optimization problem over a space of probability functions, to derive an optimal probabilistic policy $\pi$ with the Wasserstein distance as the regularization term. The objective function of $\text{RBoN}_{\text{SWD}}$ is the following:

$$\pi_{\mathrm{SRBoN_{WD}}}(x) = \arg\max_{\pi \in \Pi} \mathbb{E}_{y \sim \pi(\cdot|x)}[R(x, y)] - \beta \mathbf{WD}[\pi_{\mathbf{ref}}(\cdot \mid x) \| \pi(\cdot \mid x)]$$
$$= \arg\max_{\pi \in \Pi} f_{\mathrm{RRL}}^{\mathrm{WD}}(\pi). \tag{13}$$

### 3.2.1 Theoretical Guarantee of $\mathrm{SRBoN_{WD}}$

Similar to $\mathrm{SRBoN_{KL}}$, $\mathrm{SRBoN_{WD}}$ can also be reformulated as a max-min problem, and thus we can show that it optimizes the worst-case performance under certain constrain:

**Theorem 3.2.** *The problem of maximizing $f_{\mathrm{RRL}}^{\mathrm{WD}}(\pi)$ can be interpreted as a robust optimization problem with an adversarial perturbation $\Delta R$:*

$$\arg\max_{\pi \in \Pi} f_{\mathrm{RRL}}^{\mathrm{WD}}(\pi) = \arg\max_{\pi \in \Pi} \min_{\Delta R \in \mathcal{R}_\Delta} \mathbb{E}_{y \sim \pi(\cdot|x)}[R(x, y) - \beta \Delta R(x, y)] + \beta \sum_{y \in \mathcal{Y}_{ref}} \pi_{\mathbf{ref}}(y \mid x) \Delta R(x, y) \tag{14}$$

*where the feasible set of reward perturbations $\mathcal{R}_\Delta$ available to the adversary is bounded:*

$$\mathcal{R}_\Delta := \left\{ \Delta R \in \mathbb{R}^{\mathcal{X} \times \mathcal{Y}_{ref}} \mid |\Delta R(x, y) - \Delta R(x, y')| \le C(y, y') \quad \forall y, y' \in \mathcal{Y}_{\mathbf{ref}} \right\}, \tag{15}$$

The proof is provided in Appendix A.

This expression represents an optimization problem with strategies $\pi$ and perturbation $\Delta R$. The goal is to find the optimal strategy $\pi^*$ under the modified reward $R'$ $(= R - \beta \Delta R)$.

The intuition behind the second term $\sum_{y \in \mathcal{Y}_{\mathbf{ref}}} \pi_{\mathbf{ref}}(y \mid x) \Delta R(x, y)$ can be understood by examining $\Delta R$ constraints (Eq. (15)). While this feasible set does not explicitly constrain $\Delta R$ to avoid large values, the second term, $\min_{\Delta R} \mathbb{E}_\pi[R(x, y) - \beta \Delta R(x, y)]$, helps to avoid such huge values. Additionally, it reveals a mechanism that inherently limits the magnitude of perturbations for actions that have high probability under $\pi_{\mathbf{ref}}$ and this is consistent with the WD distance intuition.

We have analyzed the role of the regularization term for BoN sampling in the previous section 3.1.1 and section 3.2.1. Since the previous study (Jinnai et al., 2024) imposed deterministic constraints, the results are not exactly the same, but we consider that the analysis performed here helps to explain why the previous study performed better.

**Note** The feasible set of reward perturbations $\mathcal{R}_\Delta$ is bounded to be a Lipschitz continuous function with respect to a cost function $C$, which generally takes a non-negative value in applications. The perturbation behavior corresponds to the Lipschitz continuity condition, which has traditionally been well-treated in the RL community. For example, previous studies such as Rachelson & Lagoudakis (2010); Pirotta et al. (2015) considered continuous state and action spaces in RL and derived Lipschitz continuity for reward functions to aid their analysis.

## 4 Experimental Evaluation

We evaluate the performance of SRBoN compared to other text generation approaches. The datasets and models used in the experiments are all publicly available (Appendix I).

**Datasets.** We conduct experiments using two datasets: the AlpacaFarm dataset (Dubois et al., 2023) and Anthropic's hh-rlhf (HH) dataset, which we use the Harmlessness and Helpfulness subsets (Bai et al., 2022). For the AlpacaFarm dataset, we use the first 1000 entries of the train split (alpaca human preference) as the development set and the 805 entries of the evaluation split (alpaca farm evaluation) for evaluation. For Anthropic's datasets, we separately conduct experiments on the helpful-base (Helpfulness) and harmless-base (Harmlessness). For each dataset, we use the first 1000 entries of the train split as the development set and the first 1000 entries of the evaluation split for evaluation.

Table 1: Description of the text generation algorithms evaluated in the experiments. A checkmark (✓) indicates that the method uses the specified function, while a blank space means that it does not.

| Method | Reward Function | Similarity Function | Description |
|---|:---:|:---:|---|
| Random sampling | | | Use an output that is randomly sampled from the reference model. |
| Best-of-N (BoN) (Stiennon et al., 2020) | ✓ | | Generate N outputs, evaluate with reward function, select the best. |
| MBR (Eikema & Aziz, 2022) | | ✓ | Generate N outputs, evaluate with expected utility function, select the best. (Details in section 4) |
| RBoN$_{\mathrm{KL}}$ (Jinnai et al., 2024) | ✓ | | Maximize the mixture of the reward function and KL divergence with a constraint that the resulting policy is deterministic. |
| RBoN$_{\mathrm{WD}}$ (Jinnai et al., 2024) | ✓ | ✓ | Maximize the mixture of the reward function and WD distance with a constraint that the resulting policy is deterministic. |
| SRBoN$_{\mathrm{KL}}$ **(Section 3.1)** | ✓ | | Maximize the mixture of the reward function and KL divergence. |
| SRBoN$_{\mathrm{WD}}$ **(Section 3.2)** | ✓ | ✓ | Maximize the mixture of the reward function and WD distance. |
| RBoN$_{\mathrm{L}}$ **(Section 4)** | ✓ | | Consider both the reward function and the token length of the sentence. (Details in section 4 and Appendix G) |

**Language Model, Reward Model, and Embedding Model.** We employ Mistral 7B SFT $\beta$ (Jiang et al., 2023a) as the language models. We set the maximum entry length and the maximum output length to be 256 tokens. We sample response texts using nucleus sampling (Holtzman et al., 2020) with temperature set to 1.0 and top-p set to 0.9. For each entry, in the AlpacaFarm dataset and Anthropic's datasets, 128 responses are generated using Mistral 7B SFT $\beta$.

To evaluate the performance of the algorithms under different preferences, we use OASST (reward-model-deberta-v3-large-v2), SHP-Large (SteamSHP-flan-t5-large), SHP-XL (SteamSHP-flan-t5-xl), PairRM, RM-Mistral-7B and Eurus-RM-7b (Köpf et al., 2023; Ethayarajh et al., 2022; Jiang et al., 2023b; Dong et al., 2023; Yuan et al., 2024) as reward models. For the text embedding model we use all-mpnet-base-v2 (Song et al., 2020), a sentence transformer model (Reimers & Gurevych, 2019) shown to be effective in various sentence embedding and semantic search tasks.

**Baselines.** The list of text generation methods we evaluate is present in Table 1. The baseline methods include random sampling (nucleus sampling; Holtzman et al. 2020), Best-of-N (BoN) sampling, Minimum Bayes Risk (MBR) decoding, and RBoN$_{\mathrm{L}}$, which we describe in the following.

**Minimum Bayes Risk (MBR) decoding** (Kumar & Byrne, 2002; 2004; Eikema & Aziz, 2022) is a text generation strategy that selects an output from $N$ outputs that maximizes the expected utility (Berger, 1985). Let a utility function $u(h, y)$ quantify the benefit of choosing $h \in \mathcal{Y}_{\mathbf{ref}}$ if $y$ is the correct output. Then, MBR decoding is defined as follows:

$$y_{\mathrm{MBR}}(x) = \arg\max_{h \in \mathcal{Y}_{\mathbf{ref}}} \sum_{y \in \mathcal{Y}_{\mathbf{ref}}} \frac{1}{N} u(h, y). \tag{16}$$

We include MBR decoding as one of the baselines because it has been shown to be effective in a variety of text generation tasks (Suzgun et al., 2023; Bertsch et al., 2023; Li et al., 2024; Heineman et al., 2024). We

follow the implementation of Jinnai et al. (2024) and use the cosine similarity of the sentence embedding as the utility function. We use the same embedding model as the RBoN$_{\mathrm{WD}}$, all-mpnet-base-v2. Note that MBR corresponds to RBoN$_{\mathrm{WD}}$ with $u(h, y) = 1 - C(h, y)$ with no reward function or $\beta \to +\infty$ (Eq. (6)) (Jinnai et al., 2024).

As an additional evaluation method, we propose **Sentence Length Regularized BoN** (RBoN$_{\mathrm{L}}$), a simple baseline that adjusts the output token length to the target reward model. In RBoN$_{\mathrm{KL}}$ and SRBoN$_{\mathrm{KL}}$, $\pi_{\mathbf{ref}}$ was used for regularization. However, we have observed a bias in language models with respect to sentence length, namely that these models tend to produce shorter sentences with higher probability (Appendix B). To this end, we propose a simple implementation of RBoN that regularizes the generation probability of the sequence token length instead of the generation probability of each sequence. The objective function of RBoN$_{\mathrm{L}}$ is given by:

$$y_{\mathrm{RBoN_L}}(x) = \underset{y \in \mathcal{Y_{\mathbf{ref}}}}{\arg \max} \; R(x, y) - \frac{\beta}{|y|}, \tag{17}$$

where $\beta$ is a regularization parameter and $|y|$ denotes the sequence length (i.e., the number of tokens).

The rationale for this particular form of the regularization term and the experimental details of this approach are described in Appendix G.

## 4.1 Evaluation of the Algorithms

**Setup.** We compare the 7 methods using win rates vs. BoN sampling on the evaluation splits of the datasets. Since the RBoN method has a hyperparameter $\beta$, we first find the optimal $\beta^*$ on the train splits. Hyperparameter $\beta$ range is $\{1.0 \times 10^{-4}, 2.0 \times 10^{-4}, 5.0 \times 10^{-4}, 1.0 \times 10^{-3}, ..., 2.0 \times 10^1\}$. We first find the optimal beta value $\beta^*$ in the train split, then we use the optimal values in the development split for the evaluation split. In this experiment, we use OASST, SHP-Large, SHP-XL, PairRM, and RM-Mistral-7B as proxy reward models. As as the gold reward model, we use Eurus-RM-7B to evaluate the performance of the algorithms. We evaluate the performance of the algorithms as the win rate against BoN sampling according to the reward score of the gold reward model (we count ties as 0.5 wins). We use Eurus-RM-7B as the gold reward model because it is reproducible as it is open source and has been shown to have a high correlation with human preference in RewardBench (Lambert et al., 2024b).

**Results.** Table 2 reveals several noteworthy results for the AlpacaFarm, Harmlessness, and Helpfulness datasets and the optimal beta $\beta^*$ is Table 4. The win rate result shows that higher Spearman rank correlation values (Table 3) correspond to better BoN sampling accuracy. This observation is intuitive.

Table 2 shows that the win rate of SRBoN$_{\mathrm{KL}}$ is inferior to the deterministic version RBoN$_{\mathrm{KL}}$. While SRBoN$_{\mathrm{KL}}$ is proposed as a theoretically robust algorithm (section 3.2.1), its performance in our experiments did not fully meet expectations. One possible factor contributing to this discrepancy could be related to the perturbation range of $\Delta R$. In our experimental setup, it is plausible that the actual perturbations of $\Delta R$ may have exceeded the assumed theoretical limits.

Other reasons for suboptimal performance, applicable to both deterministic and stochastic versions, concern the relationship between the reference policy $\pi_{\mathbf{ref}}$ and the reward model. If the correlation between $\pi_{\mathbf{ref}}$ and the reward model is weak, the regularization effect may not contribute positively to the performance of the algorithm (Appendix B).

SRBoN$_{\mathrm{WD}}$ shows superior performance across several settings and achieves comparable performance to RBoN$_{\mathrm{WD}}$. This robust performance is remarkable given the low positive correlation between the reference policy $\pi_{\mathbf{ref}}$ and the reward model.

A plausible explanation for this effectiveness, especially in contrast to SRBoN$_{\mathrm{KL}}$, is the constraint on the reward perturbation $\Delta R$ in SRBoN$_{\mathrm{WD}}$. Unlike SRBoN$_{\mathrm{KL}}$, the constraint on $\Delta R$ in SRBoN$_{\mathrm{WD}}$ is independent of $\pi_{\mathbf{ref}}$, which mitigates low performance when there is no correlation between the reward model and $\pi_{\mathbf{ref}}$.

Table 2: The win rate of various methods against BoN sampling.

| Method | OASST | SHP-Large | SHP-XL | PairRM | RM-Mistral-7B |
|---|---|---|---|---|---|
| **AlpacaFarm** | | | | | |
| BoN | 50.0 | 50.0 | 50.0 | 50.0 | 50.0 |
| MBR | 36.0 | 42.8 | 40.8 | 39.1 | 13.0 |
| Random | 20.5 | 30.3 | 29.4 | 27.1 | 3.0 |
| $RBoN_{WD}$ | 50.6 | 50.2 | 49.0 | **50.7** | 49.9 |
| $RBoN_{KL}$ | 47.7 | 26.4 | 26.2 | 50.0 | 48.6 |
| $RBoN_L$ | **52.0** | 50.3 | **50.2** | 50.1 | **50.8** |
| $SRBoN_{WD}$ | 50.1 | **50.6** | 49.5 | 50.0 | 50.1 |
| $SRBoN_{KL}$ | 12.6 | 20.9 | 18.7 | 28.0 | 4.7 |
| | **OASST** | **SHP-Large** | **SHP-XL** | **PairRM** | **RM-Mistral-7B** |
| **Harmlessness** | | | | | |
| BoN | 50.0 | 50.0 | 50.0 | 50.0 | 50.0 |
| MBR | 40.8 | 57.4 | 50.7 | 42.7 | 14.8 |
| Random | 26.7 | 52.7 | 46.3 | 28.0 | 7.1 |
| $RBoN_{WD}$ | 52.1 | **62.2** | **57.1** | 50.0 | 49.9 |
| $RBoN_{KL}$ | 48.2 | 46.9 | 40.4 | 50.0 | 47.4 |
| $RBoN_L$ | **52.2** | 54.8 | 54.2 | 50.0 | **51.6** |
| $SRBoN_{WD}$ | 49.7 | 51.2 | 49.8 | 50.0 | 49.9 |
| $SRBoN_{KL}$ | 20.5 | 42.3 | 37.1 | 30.4 | 5.5 |
| | **OASST** | **SHP-Large** | **SHP-XL** | **PairRM** | **RM-Mistral-7B** |
| **Helpfulness** | | | | | |
| BoN | 50.0 | 50.0 | 50.0 | 50.0 | 50.0 |
| MBR | 41.4 | 39.2 | 33.2 | 40.0 | 6.1 |
| Random | 23.6 | 23.7 | 15.1 | 23.3 | 0.8 |
| $RBoN_{WD}$ | 52.5 | **52.4** | 50.1 | **50.1** | 49.9 |
| $RBoN_{KL}$ | 44.9 | 19.9 | 13.9 | 50.0 | 50.0 |
| $RBoN_L$ | **52.7** | 49.9 | **50.8** | 50.0 | **50.2** |
| $SRBoN_{WD}$ | 50.4 | 49.5 | 49.6 | 50.0 | 50.0 |
| $SRBoN_{KL}$ | 13.4 | 18.5 | 11.8 | 24.3 | 1.4 |

Table 3: Spearman's rank correlation between Eurus-RM-7B and each proxy reward. The comprehensive Spearman's rank correlation results for all the aforementioned analyses are presented in Appendix D.

| Dataset | OASST | SHP-Large | SHP-XL | PairRM | RM-Mistral-7B |
|---|---|---|---|---|---|
| AlpacaFarm | 0.39 | 0.29 | 0.35 | 0.33 | 0.62 |
| Harmlessness | 0.37 | 0.09 | 0.14 | 0.36 | 0.60 |
| Helpfulness | 0.39 | 0.38 | 0.50 | 0.34 | 0.75 |

Despite its simple implementation, $RBoN_L$ consistently outperformed BoN sampling, achieving a higher win rate on almost all tasks and models with no instances of underperformance. A detailed discussion of $RBoN_L$ is presented in Appendix G.

## 4.2 RBoN Sensitiveness of Parameters

**Setup.** In this section, we evaluate the generalization performance of the model using $\beta$. values $\{1.0 \times 10^{-4},$ $2.0 \times 10^{-4}, 5.0 \times 10^{-4}, 1.0 \times 10^{-3}, ..., 2.0 \times 10^{1}\}$ to the evaluation splits. We also use several models as proxy reward models, including OASST, SHP-Large, SHP-XL, PairRM, and RM-Mistral-7B. As a gold reward model, we use Eurus-RM-7B to evaluate the performance of the proxy models. The results are visualized as

Table 4: Optimal beta $\beta^*$ in the train split

| Method | OASST | SHP-Large | SHP-XL | PairRM | RM-Mistral-7B |
|---|---|---|---|---|---|
| **AlpacaFarm** | | | | | |
| RBoN$_\text{WD}$ | 20 | 0.5 | 0.5 | 20 | 0.1 |
| RBoN$_\text{KL}$ | 0.0001 | 0.0001 | 0.0001 | 0.0001 | 0.0001 |
| RBoN$_\text{L}$ | 20 | 0.5 | 0.2 | 20 | 15.0 |
| SRBoN$_\text{WD}$ | 0.5 | 0.0002 | 0.0001 | 0.0001 | 1.0 |
| SRBoN$_\text{KL}$ | 20 | 0.05 | 0.05 | 20 | 20 |
| **Harmlessness** | | | | | |
| RBoN$_\text{WD}$ | 20 | 1.0 | 1.0 | 0.0001 | 5.0 |
| RBoN$_\text{KL}$ | 0.0001 | 0.0001 | 0.0001 | 0.0001 | 0.0001 |
| RBoN$_\text{L}$ | 20 | 5.0 | 5.0 | 0.0001 | 20 |
| SRBoN$_\text{WD}$ | 0.05 | 0.0001 | 0.0001 | 0.0001 | 0.02 |
| SRBoN$_\text{KL}$ | 20 | 0.05 | 20 | 20 | 20 |
| **Helpfulness** | | | | | |
| RBoN$_\text{WD}$ | 15.0 | 0.05 | 0.1 | 20 | 0.5 |
| RBoN$_\text{KL}$ | 0.0001 | 0.0001 | 0.0001 | 0.0001 | 0.0001 |
| RBoN$_\text{L}$ | 20 | 0.02 | 0.2 | 5.0 | 20 |
| SRBoN$_\text{WD}$ | 0.5 | 0.001 | 0.005 | 5.0 | 0.0002 |
| SRBoN$_\text{KL}$ | 20 | 0.05 | 20 | 20 | 20 |

a plot showing the win rates of each method compared to BoN sampling on the evaluation splits. We assign 1 point for a win and 0.5 points for a tie.

**Results** The performance result of RBoN method in AlpacaFarm is illustrated in Figures 1. This result reveals that the optimal parameters for the RBoN$_\text{WD}$ and SRBoN$_\text{WD}$ method vary between different models and reveals the performance of SRBoN$_\text{WD}$ across various problem settings, as the value of the regularization parameter $\beta$ increases, we observe a degradation performance. Intuitively, upon examining the adversarial formulation of SRBoN$_\text{WD}$, we can infer that as the regularization parameter $\beta$ increases, the magnitude of potential perturbations $\Delta R$ also increases. Furthermore, as evidenced in Table 4, the optimal $\beta$ value for SRBoN$_\text{WD}$ is typically smaller than that for RBoN$_\text{WD}$.

This result shows that SRBoN$_\text{KL}$ consistently underperforms within the $\beta$ range examined in our experiments. In particular, as shown in Table 4, the optimal regularization parameter $\beta^*$ for SRBoN$_\text{KL}$ is often found to be $\beta^* = 20$ across different problem settings. This observation leads to an intriguing hypothesis, that the performance of SRBoN$_\text{KL}$ could potentially improve with higher values of $\beta$.

The performance result of RBoN$_\text{L}$ demonstrates superior performance across a wide range of $\beta$ values, exhibiting performance characteristics comparable to RBoN$_\text{WD}$. Notably, this robust performance across varying $\beta$ values indicates that RBoN$_\text{L}$ exhibits low sensitivity to changes in the regularization parameter.

The results for Harmlessness and Helpfulness datasets are presented in Appendix C.

## 5 Related Work

**Robust MDPs** Several studies have investigated RL considering the worst-case scenario for rewards. Ortega & Lee (2014) considers only a single-step analysis for the reward robust problem. Husain et al. (2021) proposes a deep RL algorithm related to Q learning for the reward robust problem. Derman et al. (2021) considers both a reward function and the transition probability as unknown. The policy regularization is considered a perturbation of the rewards, while the transition probability perturbations address the worst-case scenario with respect to the associated set of value functions. They define specific uncertainty sets and conduct thorough experiments. Eysenbach & Levine (2022) shows that incorporating the policy's Shannon

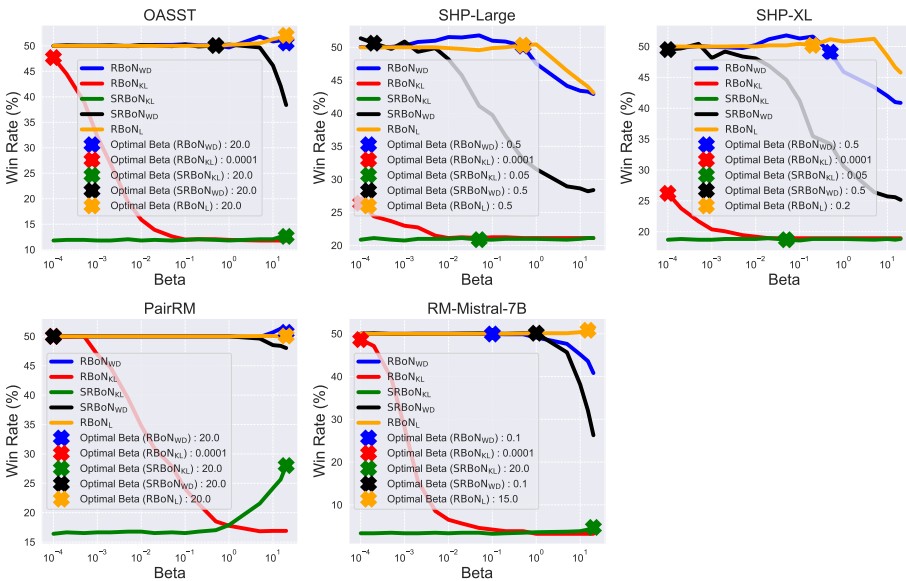

Figure 1: Evaluation of RBoN sensitiveness on the AlpacaFarm dataset with varying parameter $\beta$. We use proxy reward models, OASST, SHP-Large, SHP-XL, PairRM, and RM-Mistral-7B. As the gold reward model, we utilize Eurus-RM-7B.

entropy into the reinforcement learning objective function represents the worst-case scenario for a given uncertainty set of rewards.

**Alignment Strategies** Two notable adaptation strategies have recently gained attention: Reinforcement Learning from Human Feedback (RLHF) and Direct Preference Optimization (DPO) (Stiennon et al., 2020; Rafailov et al., 2023). RLHF incorporates human feedback into the reinforcement learning process to align the agent's behavior with human preferences. By using human feedback as a reward signal, RLHF aims to optimize it. This approach has been successfully used in LLM (Ouyang et al., 2022). On the other hand, DPO uses the same objective function as RLHF without an explicit reward function. However, it still suffers more than RLHF from overoptimization when dealing with out-of-distribution data (Xu et al., 2024). Beyond the methods discussed, there is research in robust optimization that addresses the development of robust algorithms for scenarios with unstable preference information (Wu et al., 2024). In particular, Chowdhury et al. (2024) have introduced a robust DPO approach that achieves robustness without explicitly employing robust optimization techniques. Another technique (Mudgal et al., 2024) is to train a token-level scoring value function module to select the optimal output. Khanov et al. (2024) is a novel decoding method that does not require additional learning and uses both the language model and the reward model knowledge. There is a parameter that determines which is more important, and depending on its value, it can be a conventional method.

# 6 Conclusions

This paper introduces three novel BoN sampling methods: $\text{SRBoN}_{\text{KL}}$, $\text{SRBoN}_{\text{WD}}$, and $\text{RBoN}_{\text{L}}$. To rigorously evaluate the effectiveness of these proposed methods, we conducted extensive experiments using two datasets: AlpacaFarm and Anthropic's hh-rlhf.

The $\text{SRBoN}_{\text{KL}}$ and $\text{SRBoN}_{\text{WD}}$ methods extend the previous $\text{RBoN}_{\text{KL}}$ and $\text{RBoN}_{\text{WD}}$ methods, respectively. In particular, $\text{SRBoN}_{\text{KL}}$ and $\text{SRBoN}_{\text{WD}}$ produce a stochastic optimal policy that differs from their deter-

ministic counterparts. The theoretical guarantees of their robustness increase the reliability of the methods in different scenarios.

The $\text{RBoN}_\text{L}$ method is a contribution to the field of RBoN sampling, providing a simple yet effective approach. Despite its simplicity, our experiments show that $\text{RBoN}_\text{L}$ achieves performance comparable to the more complex $\text{RBoN}_\text{WD}$. This finding highlights the potential of $\text{RBoN}_\text{L}$ as a computationally efficient alternative to more complicated methods, making it particularly attractive for applications with limited resources or stringent performance requirements.

In conclusion, this paper presents three innovative BoN sampling methods that significantly contribute to the field. The experimental results and theoretical guarantees underscore the effectiveness and reliability of these methods. Our work lays the foundation for further research and applications of robust BoN sampling techniques in a wide range of domains.

## 7 Limitations

While our proposed method demonstrates promising results, there are several limitations to note. The proposed method requires no fine-tuning of the LLMs but inevitably increases computational overhead at inference. In contrast, fine-tuning approaches incur a one-time cost during training while eliminating overhead at inference. Another concern is that the proposed method considers a max-min problem, so if, for example, the correlation between the proxy reward and the gold reward is strong, performance is reduced due to conservative output selection.

Our study lacks an analysis of whether the reward perturbations satisfy the conditions outlined in Theorems 3.1 and 3.2. Evaluating the error of the reward and utility function in experiments remains an area for future work. Additionally, the selection of the parameter $\beta$ requires a validation set in the current setting, and developing an automated method to determine $\beta$ is a promising direction for further research.

Furthermore, our approach relies on a specific utility function, which is a prerequisite for applying the proposed method, and the method does not account for process reward models, which may limit its applicability in some scenarios. It is also worth noting that the experiments conducted in this study were limited to three English datasets, leaving open the question of its generalizability to other languages or domains.

In addition, the proposed method is based on a probabilistic framework, which, while effective for uncertainty, may not align with real-world applications where deterministic versions (RBoN) are often preferred for their predictability and safety. Based on the analysis in this paper, the analysis of the deterministic RBoN is a possible direction for future work.

Finally, while the current formulation is specific, the proposed method has the potential to be extended to other divergence measures, such as $f$-divergences, offering an exciting avenue for future investigation.

## Acknowledgments

We sincerely thank the Action Editor, Pascal Poupart, and the anonymous reviewers for their insightful comments and suggestions. Kaito Ariu's research is supported by JSPS KAKENHI Grant No. 23K19986.

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

# A    Detailed proof of Theorem 3.2

**Definition A.1** (Similarity-based Lipschitz Continuity)**.** *A function f is said to have Similarity-based Lipschitz Continuity if, for any $y, y' \in \mathcal{Y}$, the following holds:*

$$|f(y) - f(y')| \leq C(y, y')$$

*where*

$$C(y, y') = 1 - \cos(\text{emb}(y), \text{emb}(y'))$$

We first explain how the objective function is reformulated to a max-min problem. Let us focus on the regularization term, 1-**WD** term rewrite related to $\pi$, $\pi_{\mathbf{ref}}$

The following analysis is done in the framework of finite probability spaces. To simplify the following proof, we introduce the following notation. Let $x_1, x_2, \cdots, x_n$ be $n$ places and consider the function $f$, where $f_i$ refers to the value $f(x_i)$.

$$\mathbf{WD}[\nu\|\mu] = \min_{\gamma \in \Gamma(\nu,\mu)} \sum_{(i,j) \in \mathcal{Y} \times \mathcal{Y}} C_{ij} \gamma_{ij}$$

$$= \min_{\gamma \in \mathbb{R}^{Y \times Y'}} \sum_{(i,j) \in \mathcal{Y} \times \mathcal{Y}} C_{ij} \gamma_{ij} + \Psi(\gamma), \tag{18}$$

where $\gamma$ is a coupling of the probability measure $\nu$ and $\mu$, $\Gamma(\nu,\mu) = \left\{ \gamma \in \mathbb{R}^{Y \times Y'} | \sum_{j \in \mathcal{Y}} \gamma_{ij} = \nu_i, \sum_{i \in \mathcal{Y}}, \gamma_{ij} = \mu_j, \gamma_{ij} \geq 0 \text{ for all } i, j \right\}$, $Y$ and $Y' \ (= \mathcal{Y})$ is sample space respectively, corresponding to outcomes $i$ and $j$ respectively and $\Psi(\gamma) = 0$ if $\gamma \in \Gamma(\nu,\mu), +\infty$ otherwise.

Constraint terms, a coupling of the probability measure $\gamma$ needs to satisfy:

$$\sum_j \gamma_{ij} = \nu_i \quad \forall i \in \mathcal{Y}$$

$$\sum_i \gamma_{ij} = \mu_j \quad \forall j \in \mathcal{Y} \tag{19}$$

$$\gamma_{ij} \geq 0 \quad \forall i, j \in \mathcal{Y}$$

This constraint can be expressed in $\mathbf{A}\gamma = \mathbf{b}$, indicating its linear nature. Specifically, $\mathbf{A}$ and $\mathbf{b}$ are defined as $\mathbf{A} = \begin{pmatrix} I_i \otimes \mathbf{1}_j^\top \\ I_j \otimes \mathbf{1}_i^\top \end{pmatrix}$, $\mathbf{b} = \begin{pmatrix} \nu \\ \mu \end{pmatrix}$. In this formulation, $I_i$ and $I_j$ denote identity matrices of dimension $\mathcal{Y} \times \mathcal{Y}$, while $\mathbf{1}_i$ and $\mathbf{1}_j$ represent column vectors of dimension $\mathcal{Y}$ with all components equal to 1. The symbol $\otimes$ denotes the Kronecker product.

**Lemma A.2.** *Eq. (18) is reformulated as a max problem from a min problem.*

$$\max_{\substack{f \\ |f_i - f_j| \leq C_{ij}}} \sum_i f_i \nu_i - \sum_j f_j \mu_j \tag{20}$$

*Proof.* Taking into account the constraints specified in Eq. (19), we proceed with the application of the Lagrange multiplier method:

$$\mathbf{WD}[\nu\|\mu] = \min_{\gamma \in \mathbb{R}^{Y \times Y'}} \sum_{i,j} C_{ij} \gamma_{ij} + \max_{f,g} \left\{ \sum_i f_i \nu_i + \sum_j g_j \mu_j - \sum_{i,j} (f_i + g_j) \gamma_{ij} \right\}$$

For a more intuitive understanding, $f$ and $g$ can be considered analogous to Lagrange multipliers. Except for the first term, all subsequent entries refer to constraints on $\gamma$.

$$\mathbf{WD}[\nu\|\mu] = \min_{\gamma \in \mathbb{R}^{Y \times Y'}} \max_{f,g} \sum_{i,j} (C_{ij} - f_i - g_j) \gamma_{ij} + \sum_i f_i \nu_i + \sum_j g_j \mu_j$$

can be seen from Eq. (19), these constraints are linear. From Theorem 5.2 (Vanderbei, 2020), in linear programming, there is never a gap between the primal and the dual optimal objective values. Under the strong duality theorem (e.g., $\min_x \max_y f(x,y) = \max_y \min_x f(x,y)$), we can exchange the min max term.

$$\mathbf{WD}[\nu\|\mu] = \max_{f,g} \min_{\gamma \in \mathbb{R}^{Y \times Y'}} \sum_{i,j}(C_{ij} - f_i - g_j)\gamma_{ij} + \sum_i f_i\nu_i + \sum_j g_j\mu_j$$

If $C_{ij} - f_i - g_j \geq 0$ for all $i,j$, the optimal value of $\min_\gamma \sum_{i,j}(C_{ij} - f_i - g_j)\gamma_{ij}$ is 0, otherwise $\infty$. This observation allows us to derive the inequality constraint for the first item. We can include this as a constraint in the equation:

$$\mathbf{WD}[\nu\|\mu] = \max_{\substack{f,g \\ f_i + g_j \leq C_{ij}}} \sum_i f_i\nu_i + \sum_j g_j\mu_j$$

Our next goal is to express the above function, currently represented by $f$ and $g$, exclusively in terms of the function $f$. From the given constraints, we have established that $f_i + g_j \leq C_{ij}$ for all $i$ and $j$. We can express this as follows:

$$g_j \leq \min_i \{C_{ij} - f_i\} \tag{21}$$

To fix $i = i^*$, since $\min_i$ picks the minimum value. The index $i^*$ gives this minimum, and fixing $i$ to $i^*$ turns the inequality in Eq. (21) into the equality in Eq. (22).

$$g_j = \{C_{i^*j} - f_{i^*}\} \tag{22}$$

Eq. (22) gives us a function which is called the $c$-transform of $f_j$ and is often denoted by $f_j^c$,

$$f_j^c = g_j = \{C_{i^*j} - f_{i^*}\}$$

We can now rewrite $\mathbf{WD}$ with $f_j^c$ as

$$\mathbf{WD}[\nu\|\mu] = \max_f \sum_i f_i\nu_i + \sum_j f_j^c\mu_j \tag{23}$$

If $f$ is similarity-based Lipschitz, $f^c$ is also similarity-based Lipschitz, for all $\boldsymbol{i}$ and $\boldsymbol{j}$ we have

$$\left|f_j^c - f_i^c\right| \leq C_{ij}$$
$$\implies -C_{ij} \leq f_j^c - f_i^c \leq C_{ij}$$
$$\implies -f_i^c \leq C_{ij} - f_j^c$$

$$\implies -f_i^c \leq \min_j \{C_{ij} - f_j^c\}$$

Upper bound of $\min_j \{C_{ij} - f_j^c\}$ is choosing $j \to i$

$$\min_j \{C_{ij} - f_j^c\} \leq -f_i^c$$

It can be shown that $f_i^{cc} = f_i = \min_j \{C_{ij} - f_j^c\}$. This means that $-g = -f^c = f$. Substituting $f_j^c = -f_j$ into Eq. 23, we get

$$\max_{\substack{f \\ |f_i - f_j| \leq C_{ij}}} \sum_i f_i\nu_i - \sum_j f_j\mu_j \tag{24}$$

which is the dual form of 1-Wasserstein distance.

$\square$

Finally, by substituting $\Delta R$ for $f$, we get:

$$\pi_{\mathrm{SRBoN_{WD}}}(x) = \max_{\pi \in \Pi} \mathbb{E}_{y \sim \pi(\cdot|x)}[R(x,y)] - \Omega(\pi)$$

$$= \max_{\pi \in \Pi} \mathbb{E}_{y \sim \pi(\cdot|x)}[R(x,y)] - \max_{\Delta R \in \mathcal{R}_\Delta} \beta \left( \sum_{\mathcal{Y}_{\mathbf{ref}}} \Delta R(x,y) \pi_{\mathbf{ref}}(y \mid x) - \sum_{\mathcal{Y}_{\mathbf{ref}}} \Delta R(x,y) \pi(y \mid x) \right)$$

$$= \max_{\pi \in \Pi} \mathbb{E}_{y \sim \pi(\cdot|x)}[R(x,y)] - \min_{\Delta R \in \mathcal{R}_\Delta} \beta \left( - \sum_{\mathcal{Y}_{\mathbf{ref}}} \Delta R(x,y) \pi_{\mathbf{ref}}(y \mid x) + \sum_{\mathcal{Y}_{\mathbf{ref}}} \Delta R(x,y) \pi(y \mid x) \right)$$

where $\Omega(\pi) = \beta \mathbf{WD}[\pi_{\mathbf{ref}}(\cdot \mid x) \| \pi(\cdot \mid x)]$.

$$\pi_{\mathrm{SRBoN_{WD}}}(x) = \max_{\pi \in \Pi} \min_{\Delta R \in \mathcal{R}_\Delta} \mathbb{E}_{y \sim \pi(\cdot|x)} [R(x,y) - \beta \Delta R(x,y)] + \beta \sum_{y \in \mathcal{Y}_{\mathbf{ref}}} \pi_{\mathbf{ref}}(y \mid x) \Delta R(x,y)$$

$$\text{where} \quad \mathcal{R}_\Delta := \left\{ \Delta R \in \mathbb{R}^{\mathcal{X} \times \mathcal{Y}_{\mathbf{ref}}} \mid |\Delta R(x,y) - \Delta R(x,y')| \le C(y,y') \quad \forall y, y' \in \mathcal{Y}_{\mathbf{ref}} \right\}$$

# B Relationship Between $\pi_{\mathbf{ref}}$ and the Proxy Reward Model

Despite the theoretical robustness of $\mathrm{SRBoN_{KL}}$ demonstrated in the analyses presented in section 3.1.1, the experimental results (section 4.1 and section 4.2) did not show comparable robustness. This section aims to explain the reasons for this discrepancy. Recall the objective function of $\mathrm{SRBoN_{KL}}$:

$$\pi_{\mathrm{SRBoN_{KL}}}(x) = \max_{\pi} \mathbb{E}_{y \sim \pi(\cdot|x)}[R(x,y)] - \Omega(\pi)$$

$$= \max_{\pi} \mathbb{E}_{y \sim \pi(\cdot|x)}[R(x,y)] - \sum_{\mathcal{Y}_{\mathbf{ref}}} \pi(y \mid x) \log \frac{\pi(y \mid x)}{\pi_{\mathbf{ref}}(y \mid x)}$$

This implies that ideally, $\pi_{\mathbf{ref}}$ and the reward function $R$ should have some form of relationship (e.g. positive correlation) that facilitates learning. However, $\pi_{\mathbf{ref}}$ is influenced by complex factors such as length bias.

To verify this hypothesis, we examine two aspects: (1) the correlation between the Eurus-RM-7B reward values, which were used as the gold reward model in our experiments, and the probabilities assigned by $\pi_{\mathbf{ref}}$; (2) the relationship between the length of the outputs generated by $\pi_{\mathbf{ref}}$ and the generation probabilities of those outputs.

Table 5: The correlation between the Eurus-RM-7B reward values and the probabilities assigned by $\pi_{\mathbf{ref}}$

| AlpacaFarm | Harmlessness | Helpfulness |
|:---:|:---:|:---:|
| $-0.224$ | $0.088$ | $-0.097$ |

Table 6: The relationship between the length of the outputs generated by $\pi_{\mathbf{ref}}$ and the generation probabilities of these outputs.

| AlpacaFarm | Harmlessness | Helpfulness |
|:---:|:---:|:---:|
| $-0.877$ | $-0.924$ | $-0.854$ |

As can be seen from Table 5, there is negligible correlation between $\pi_{\mathbf{ref}}$ and Eurus-RM-7B (gold reward model) in terms of Harmlessness and Helpfulness. In addition, the domain of the AlpacaFarm dataset tends to be negatively correlated.

These results explain the performance degradation observed when this relationship is included in the regularization term. Table 6 shows that $\pi_{\mathbf{ref}}$ has a bias towards shorter sentences, with output probabilities increasing as sentence length decreases.

# C   Supplemently Results

Figures 2 and 3 show evaluation of RBoN sensitivity on the Harmlessness subset and Helpfulness of the hh-rlhf dataset. These results were similar to those seen in AlpacaFarm using section 4.2. This means that each method is not necessarily dependent on the dataset.

Figures 4 to 6 compare RBoN$_{WD}$ and SRBoN$_{WD}$ and Figures 7 to 9 compare RBoN$_{KL}$ and SRBoN$_{KL}$. These results show that SRBoN is not superior to RBoN. This is for reasons also discussed in section 4

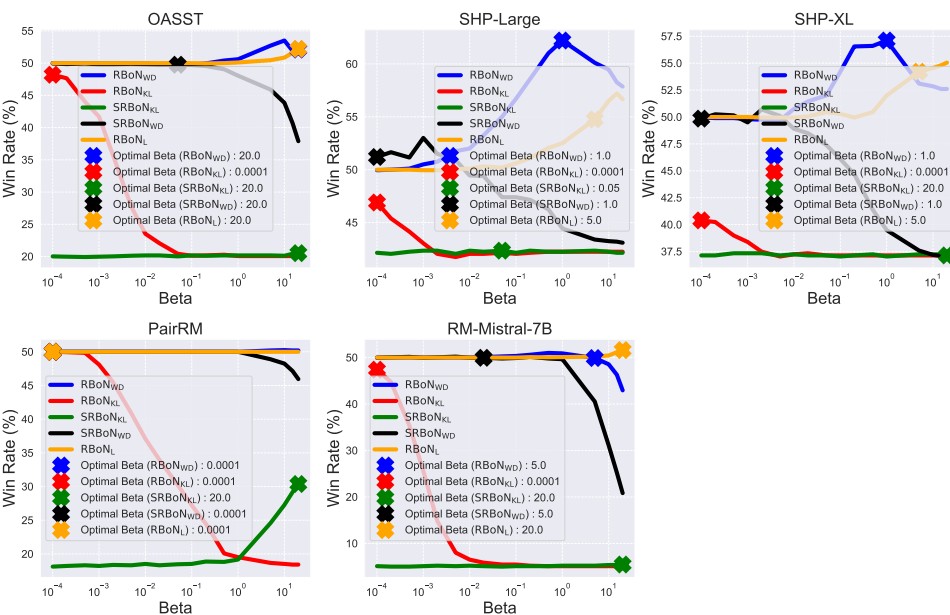

Figure 2: Evaluation of RBoN sensitiveness on the Harmlessness subset of the hh-rlhf dataset with varying parameter $\beta$. We use proxy reward models, OASST, SHP-Large, SHP-XL, PairRM, and RM-Mistral-7B. As the gold reward model, we utilize Eurus-RM-7B.

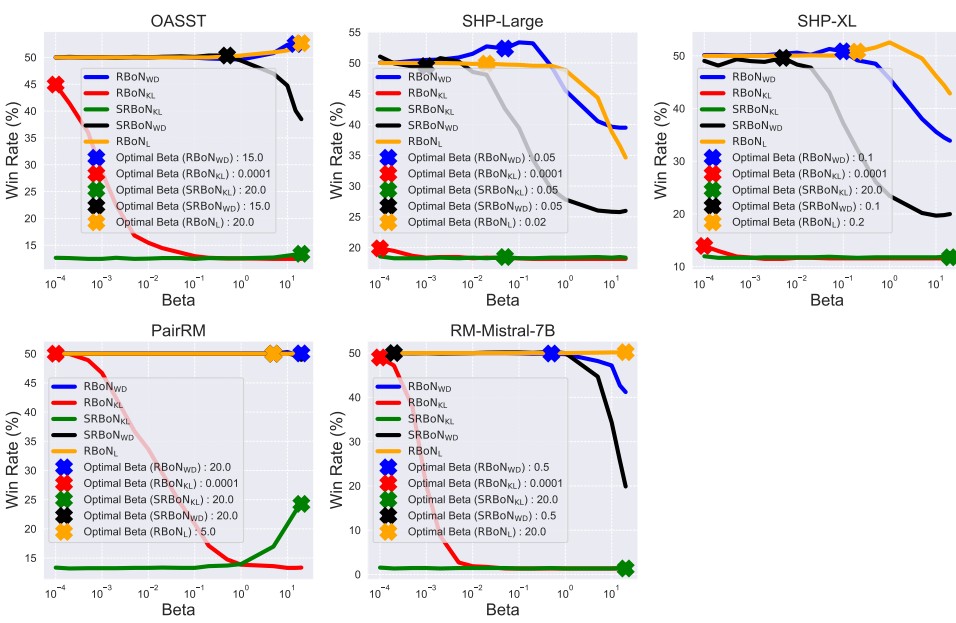

Figure 3: Evaluation of RBoN sensitiveness on the Helpfulness subset of the hh-rlhf dataset with varying parameter $\beta$. We use proxy reward models, OASST, SHP-Large, SHP-XL, PairRM, and RM-Mistral-7B. As the gold reward model, we utilize Eurus-RM-7B.

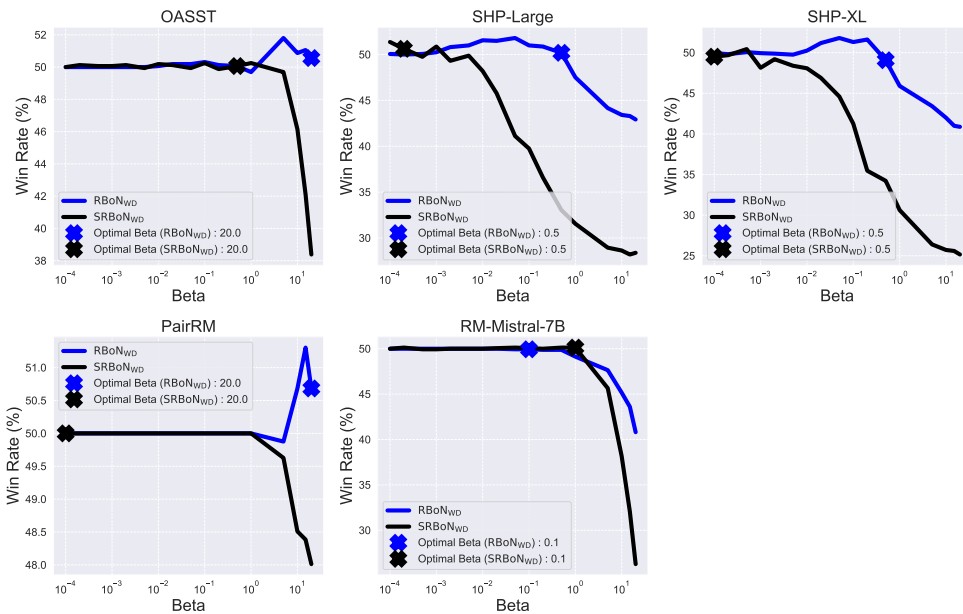

Figure 4: Evaluation of $\text{RBoN}_{\text{WD}}$ and $\text{SRBoN}_{\text{WD}}$ sensitiveness on the AlpacaFarm dataset with varying parameter $\beta$. We use proxy reward models, OASST, SHP-Large, SHP-XL, PairRM, and RM-Mistral-7B. As the gold reward model, we utilize Eurus-RM-7B.

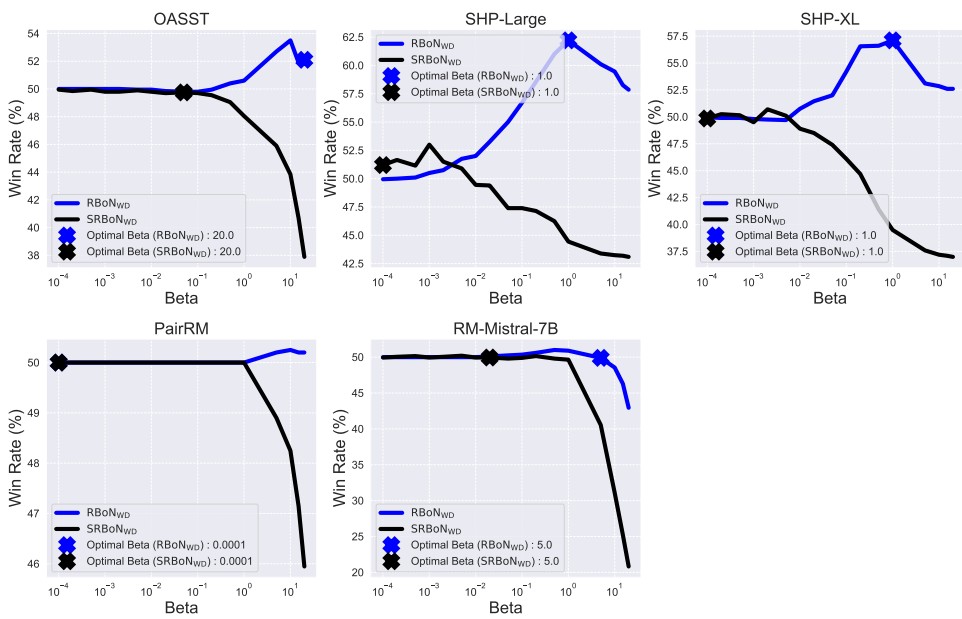

Figure 5: Evaluation of RBoN$_{\text{WD}}$ and SRBoN$_{\text{WD}}$ sensitiveness on the Harmlessness subset of the hh-rlhf dataset with varying parameter $\beta$. We use proxy reward models, OASST, SHP-Large, SHP-XL, PairRM, and RM-Mistral-7B. As the gold reward model, we utilize Eurus-RM-7B.

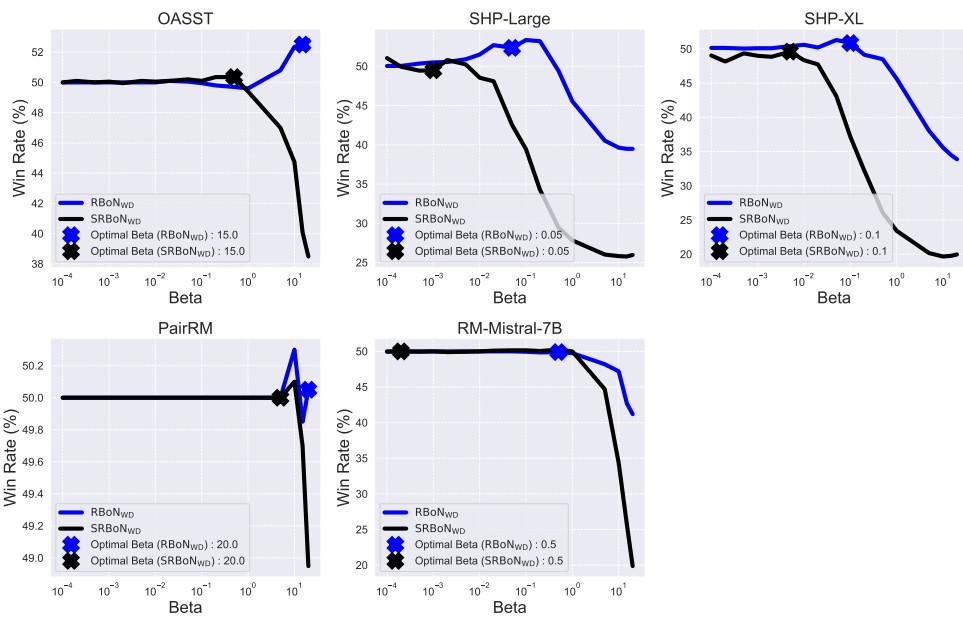

Figure 6: Evaluation of RBoN$_{\text{WD}}$ and SRBoN$_{\text{WD}}$ sensitiveness on the Helpfulness subset of the hh-rlhf dataset with varying parameter $\beta$. We use proxy reward models, OASST, SHP-Large, SHP-XL, PairRM, and RM-Mistral-7B. As the gold reward model, we utilize Eurus-RM-7B.

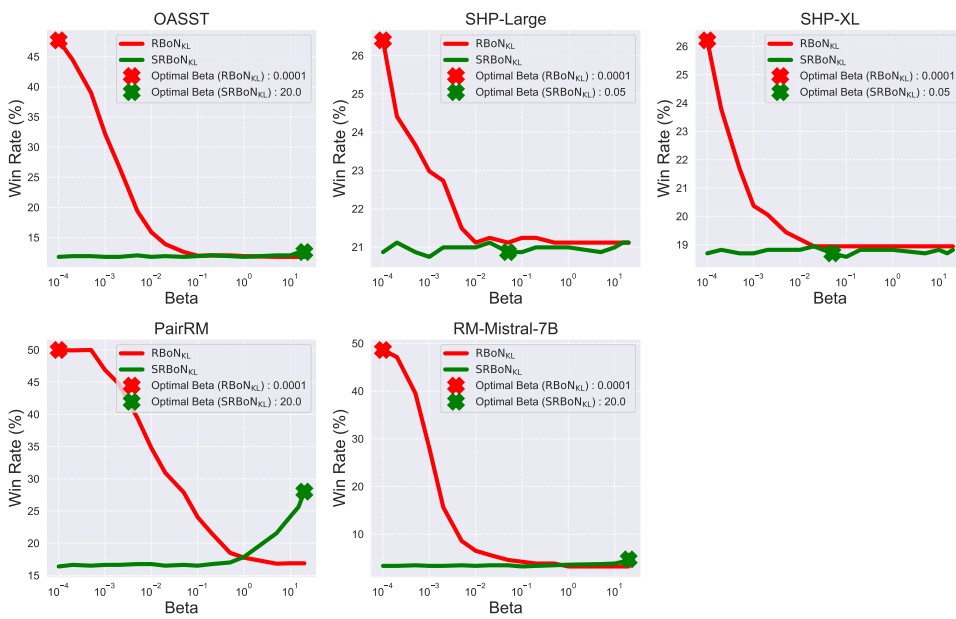

Figure 7: Evaluation of RBoN$_{KL}$ and SRBoN$_{KL}$ sensitiveness on the AlpacaFarm dataset with varying parameter $\beta$. We use proxy reward models, OASST, SHP-Large, SHP-XL, PairRM, and RM-Mistral-7B. As the gold reward model, we utilize Eurus-RM-7B.

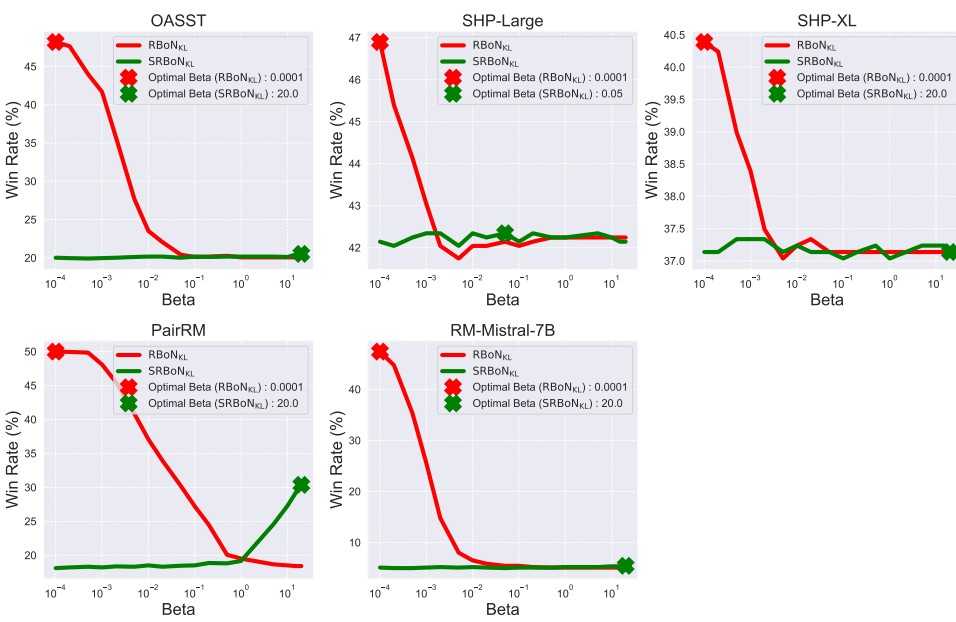

Figure 8: Evaluation of RBoN$_{KL}$ and SRBoN$_{KL}$ sensitiveness on the Harmlessness subset of the hh-rlhf dataset with varying parameter $\beta$. We use proxy reward models, OASST, SHP-Large, SHP-XL, PairRM, and RM-Mistral-7B. As the gold reward model, we utilize Eurus-RM-7B.

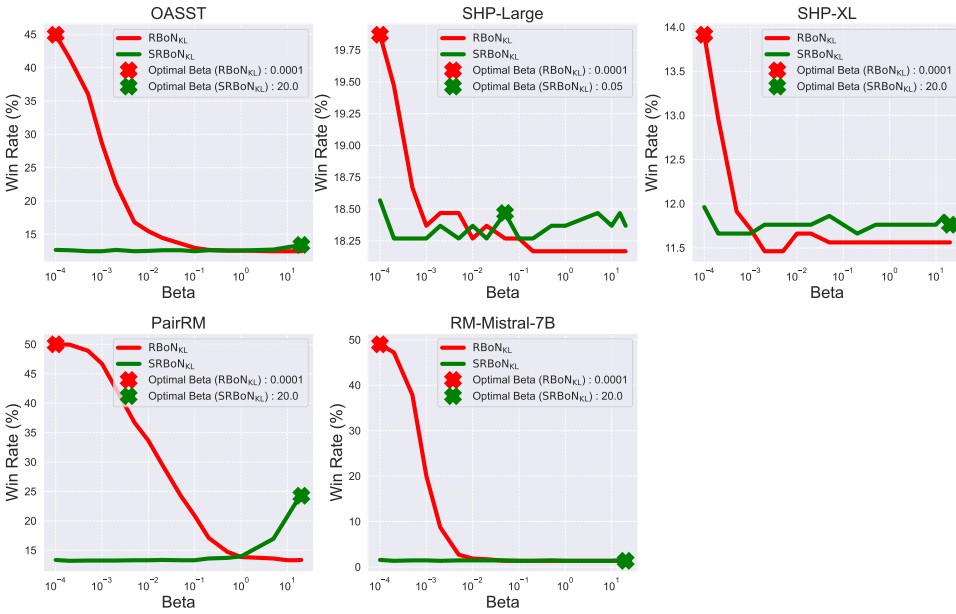

Figure 9: Evaluation of $\text{RBoN}_{KL}$ and $\text{SRBoN}_{KL}$ sensitiveness on the Helpfulness subset of the hh-rlhf dataset with varying parameter $\beta$. We use proxy reward models, OASST, SHP-Large, SHP-XL, PairRM, and RM-Mistral-7B. As the gold reward model, we utilize Eurus-RM-7B.

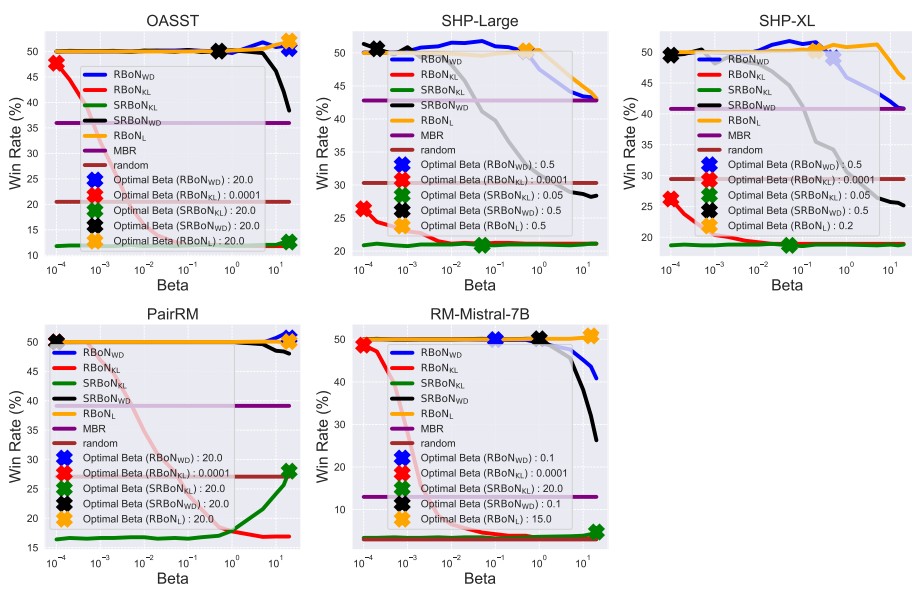

Figure 10: Evaluation of the decoder method on the AlpacaFarm dataset with varying parameter $\beta$. We use proxy reward models, OASST, SHP-Large, SHP-XL, PairRM, and RM-Mistral-7B. As the gold reward model, we utilize Eurus-RM-7B.

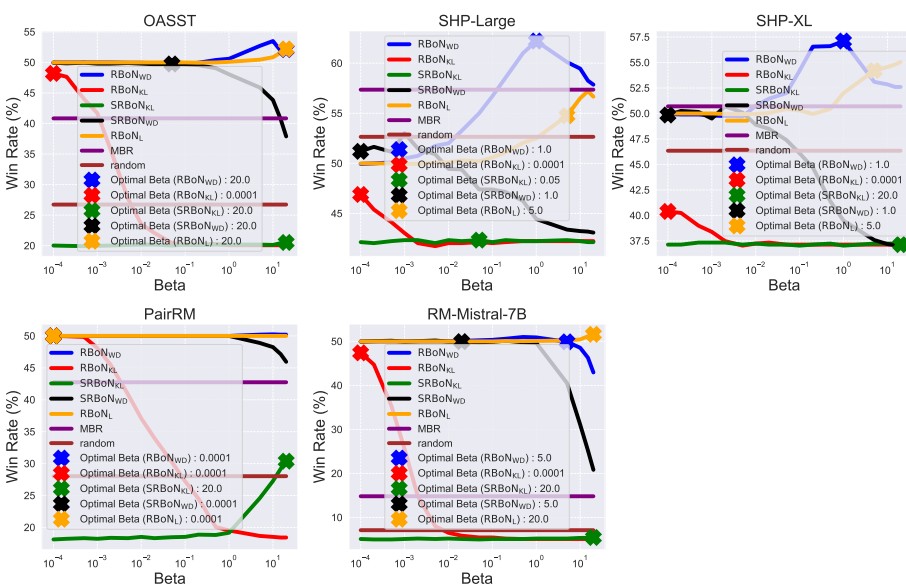

Figure 11: Evaluation of the decoder method on the Harmlessness dataset with varying parameter $\beta$. We use proxy reward models, OASST, SHP-Large, SHP-XL, PairRM, and RM-Mistral-7B. As the gold reward model, we utilize Eurus-RM-7B.

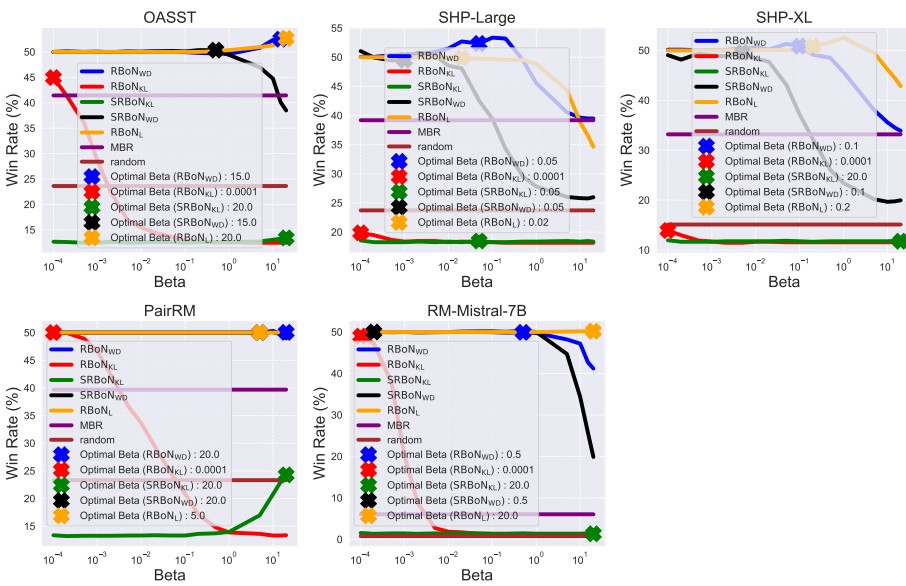

Figure 12: Evaluation of the decoder method on the Helpfulness dataset with varying parameter $\beta$. We use proxy reward models, OASST, SHP-Large, SHP-XL, PairRM, and RM-Mistral-7B. As the gold reward model, we utilize Eurus-RM-7B.

# D  Spearman's Rank Correlation (Spearman, 1904)

Figure 13, Figure 14, and Figure 15 show the average Spearman's rank correlation coefficient ($\rho$) between pairs of reward models (Spearman, 1904). These results suggest that pairs of reward models with higher correlation values are more similar, indicating a preference for greedy methods in such cases.

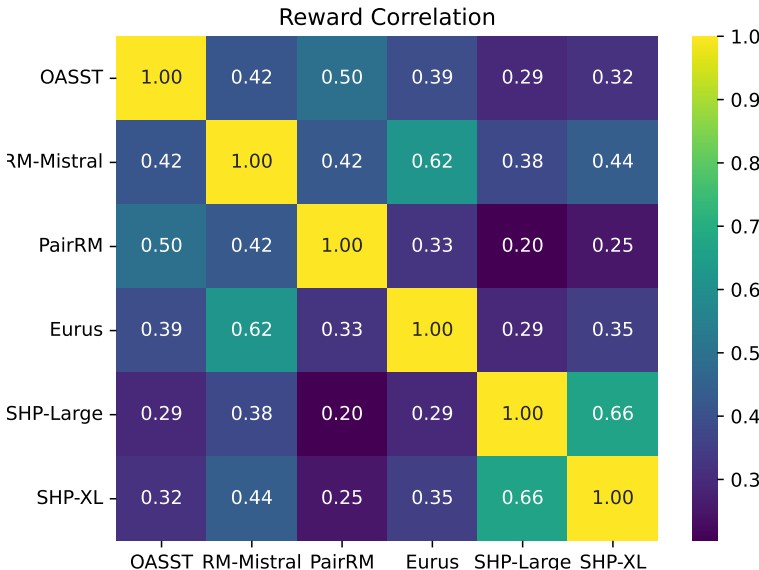

Figure 13: The average Spearman's rank correlation coefficient ($\rho$) between pairs of reward models in the AlpacaFarm dataset.

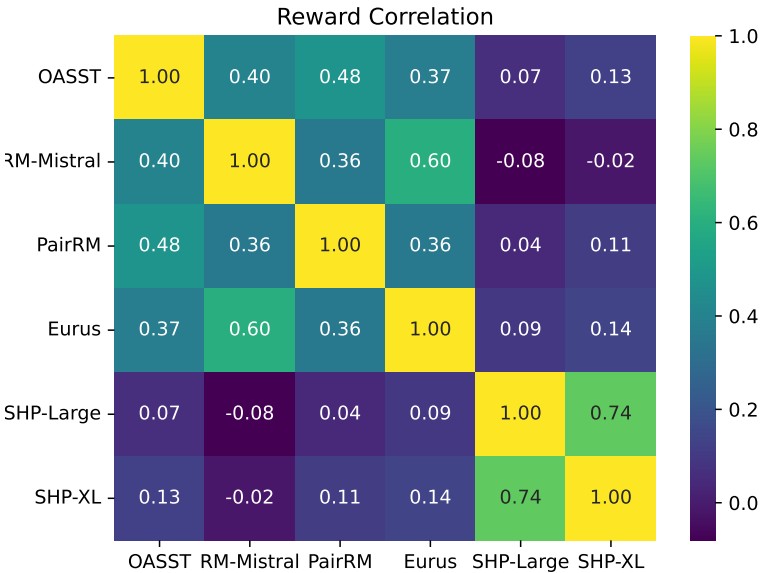

Figure 14: The average Spearman's rank correlation coefficient ($\rho$) between pairs of reward models in the Harmlessness dataset.

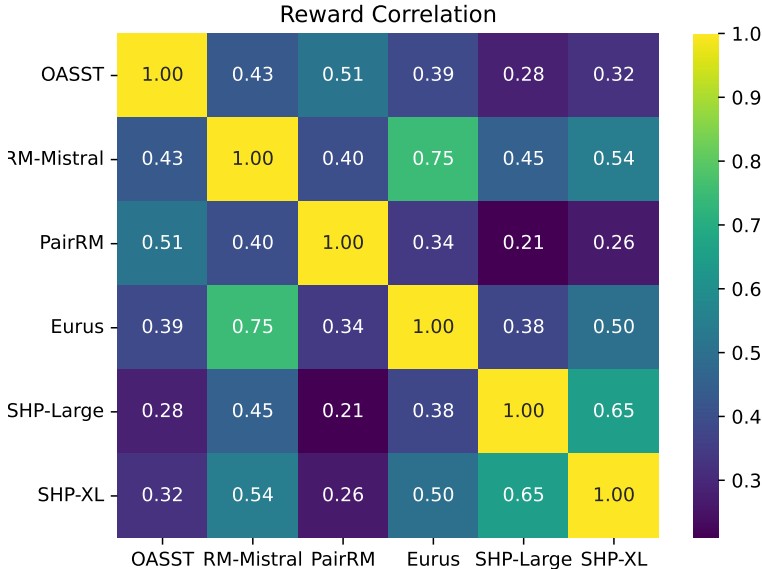

Figure 15: The average Spearman's rank correlation coefficient ($\rho$) between pairs of reward models in the Helpfulness dataset.

# E    Supplementary Result on Meta-Llama-3-8B-Instruct (Dubey et al., 2024)

We compared the average Spearman's rank correlation coefficient of the reward model and the performance of RBoN$_{\mathrm{WD}}$ on the evaluation split using the Llama (Meta-Llama-3-8B-Instruct) language model. The purpose of this analysis is to verify the performance of RBoN$_{\mathrm{WD}}$, even when applied to samples generated by state-of-the-art language models.

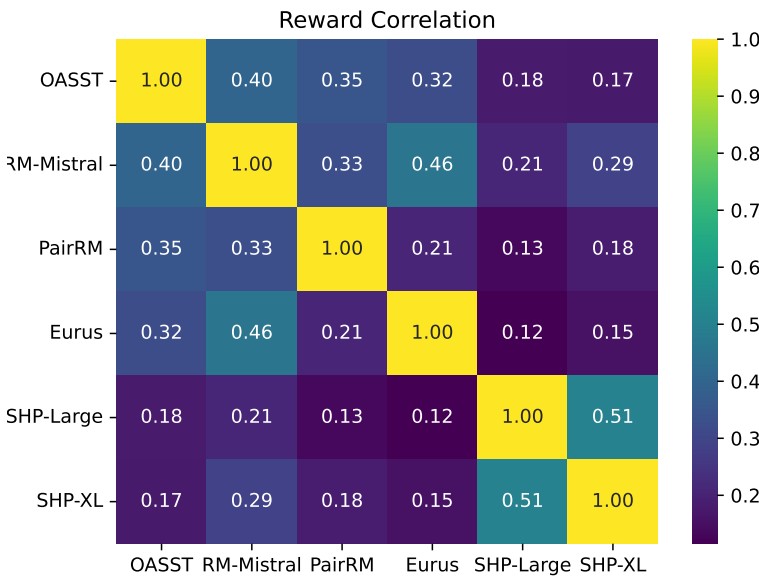

Figure 16: The average Spearman's rank correlation coefficient ($\rho$) between pairs of reward models in the AlpacaFarm dataset, using Llama as the language model.

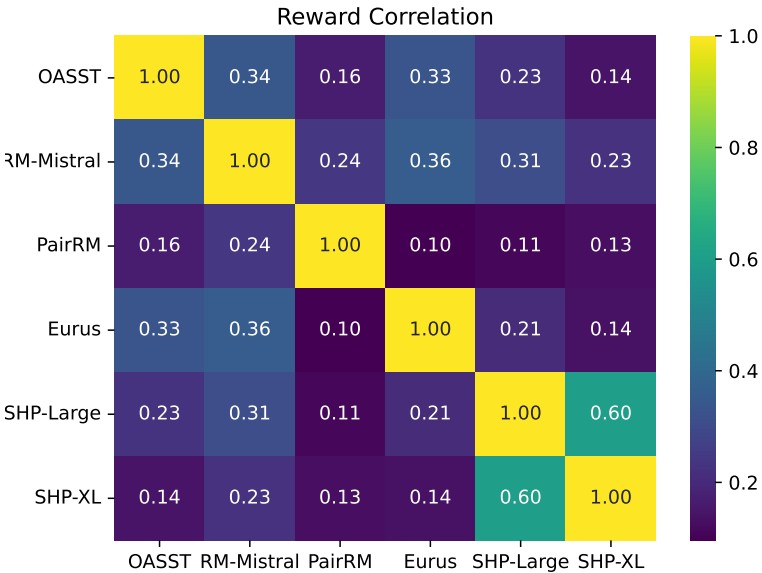

Figure 17: The average Spearman's rank correlation coefficient ($\rho$) between pairs of reward models in the Harmlessness dataset, using Llama as the language model.

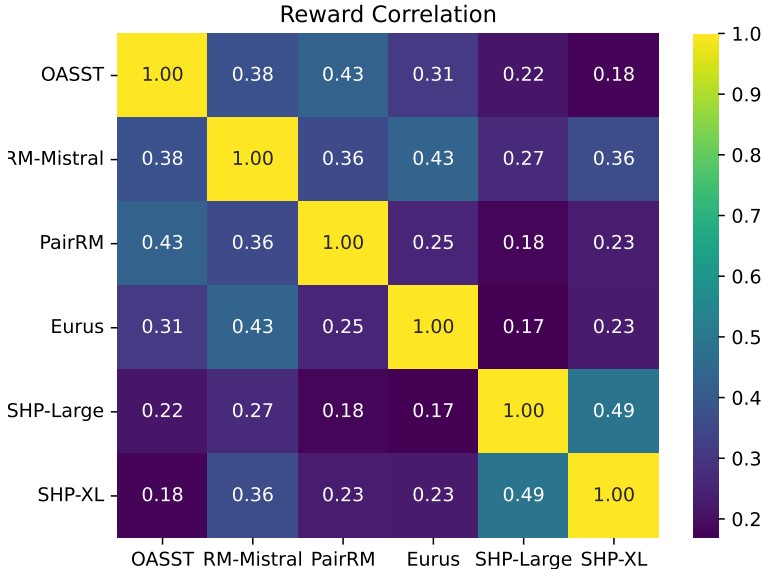

Figure 18: The average Spearman's rank correlation coefficient ($\rho$) between pairs of reward models in the Helpfulness dataset, using Llama as the language model.

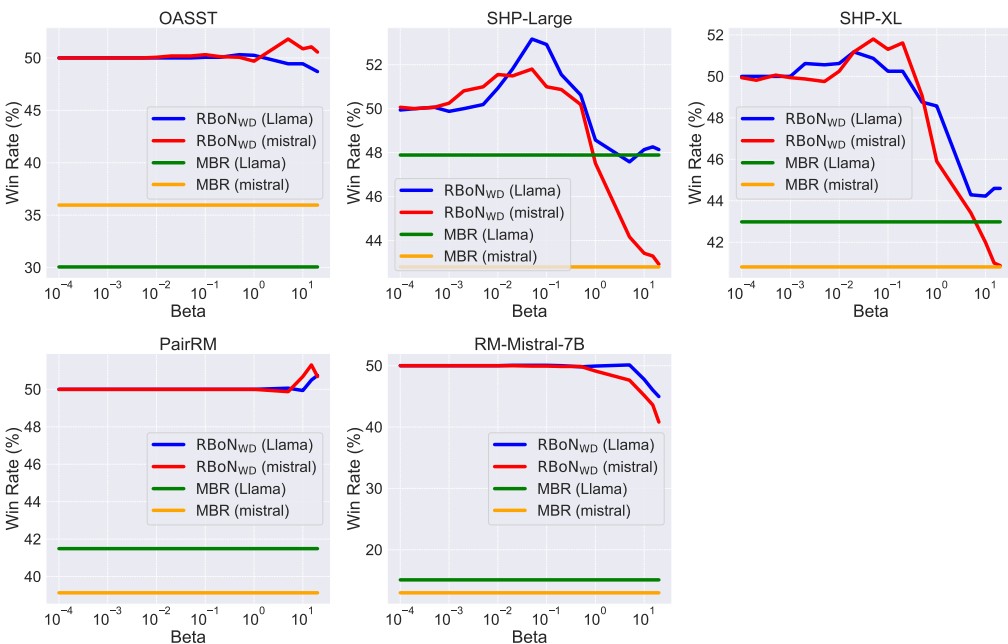

Figure 19: Evaluation of the RBoN method on the AlpacaFarm dataset with varying parameter $\beta$. We use proxy reward models, OASST, SHP-Large, SHP-XL, PairRM, and RM-Mistral-7B. As the gold reward model, we utilize Eurus-RM-7B, and Llama as the language model.

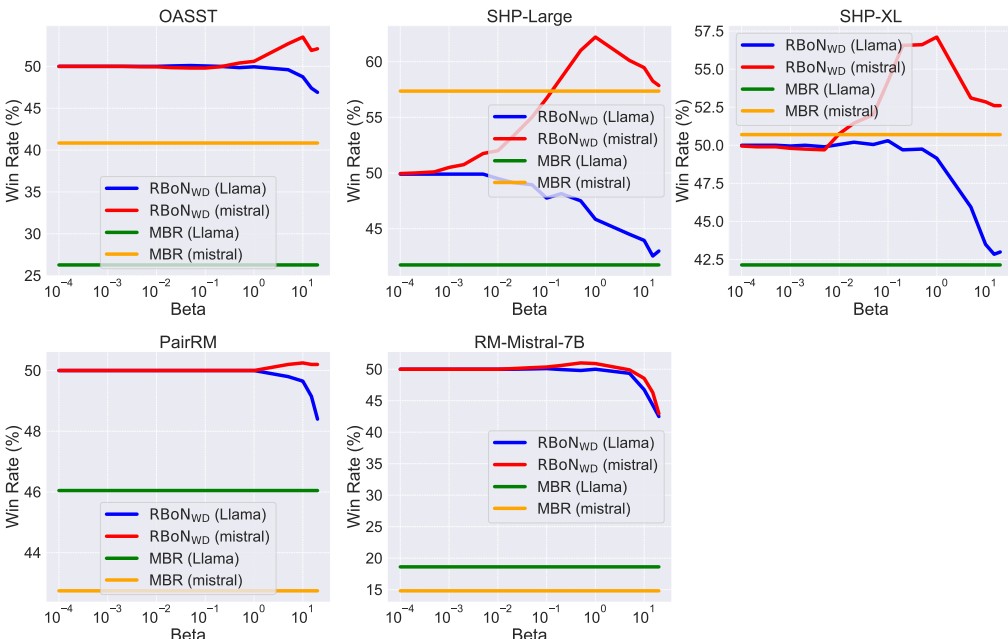

Figure 20: Evaluation of the RBoN method on the Harmlessness dataset with varying parameter $\beta$. We use proxy reward models, OASST, SHP-Large, SHP-XL, PairRM, and RM-Mistral-7B. As the gold reward model, we utilize Eurus-RM-7B, and Llama as the language model.

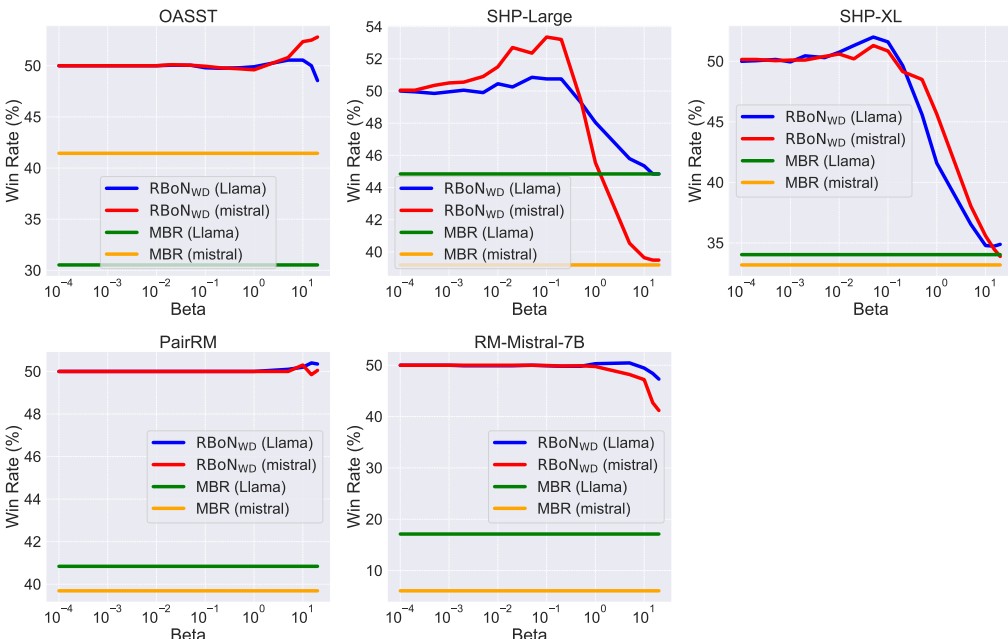

Figure 21: Evaluation of the RBoN method on the Helpfulness dataset with varying parameter $\beta$. We use proxy reward models, OASST, SHP-Large, SHP-XL, PairRM, and RM-Mistral-7B. As the gold reward model, we utilize Eurus-RM-7B, and Llama as the language model.

# F   Robustness of RBoN Under Suboptimal Reward Models

We evaluate the performance of suboptimal reward models, Beaver (beaver-7b-v1.0-reward) (Dai et al., 2024), Open Llama (hh-rlhf-rm-open-llama 3b) (Diao et al., 2024), and Tulu (tulu-v2.5-13b-uf-rm) (Ivison et al., 2024) selected from Lambert et al. (2024a), which underperforms compared to other reward models in some cases. We set these models as proxy models, set Eurus-RM-7B (Eurus) as the gold model, and also show the reward correlation of these models.

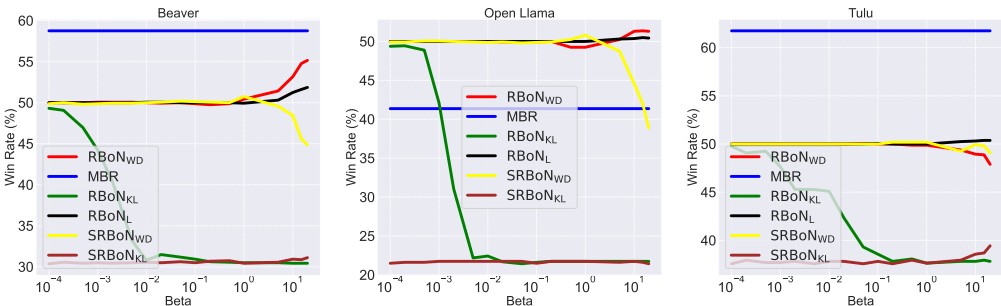

Figure 22: Evaluation of RBoN sensitiveness on the AlpacaFarm dataset with varying parameter $\beta$. We use proxy reward models, Beaver, Open Llama, and Tulu. As the gold reward model, we utilize Eurus.

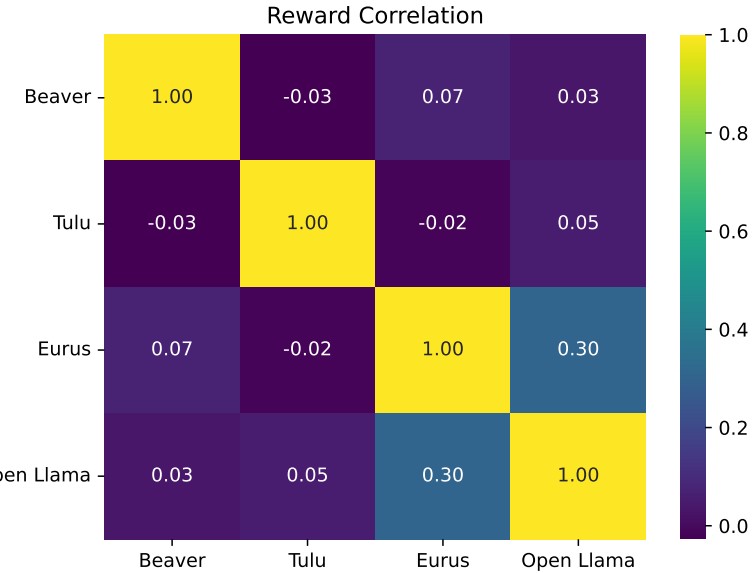

Figure 23: The average Spearman's rank correlation coefficient ($\rho$) between pairs of reward models in the AlpacaFarm dataset.

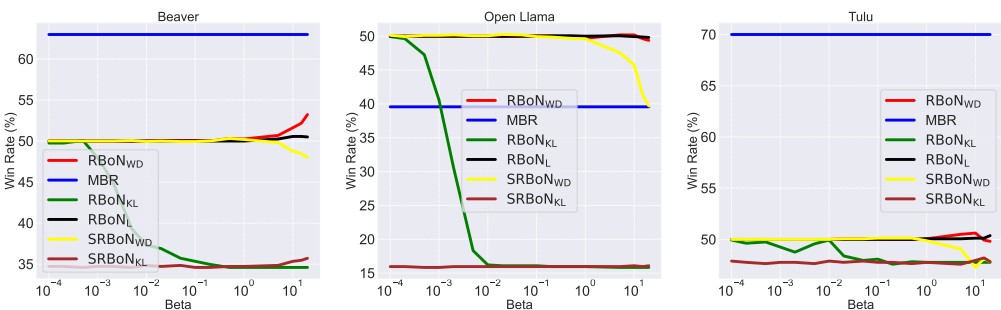

Figure 24: Evaluation of RBoN sensitiveness on the Helpfulness dataset with varying parameter $\beta$. We use proxy reward models, Beaver, Open Llama, and Tulu. As the gold reward model, we utilize Eurus.

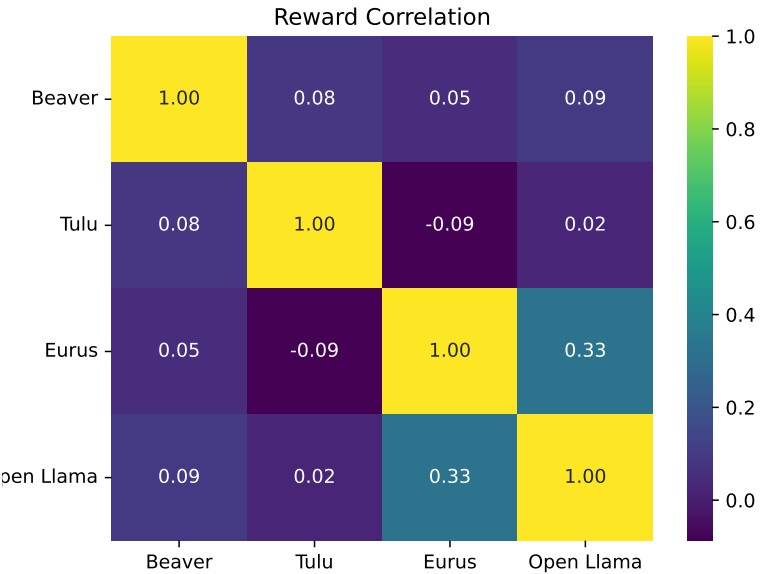

Figure 25: The average Spearman's rank correlation coefficient ($\rho$) between pairs of reward models in the Helpfulness dataset.

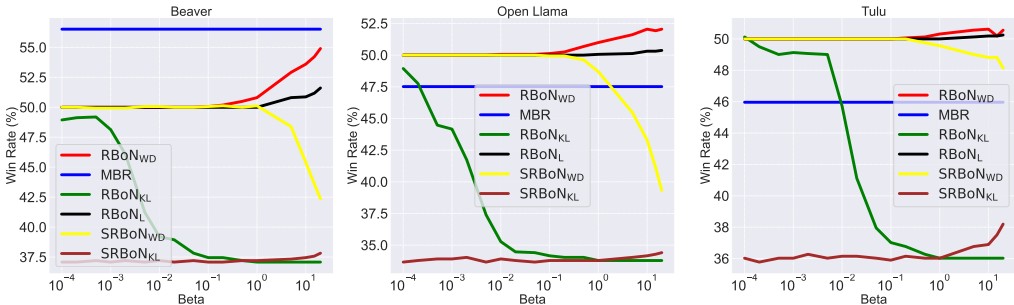

Figure 26: Evaluation of RBoN sensitiveness on the Harmlessness dataset with varying parameter $\beta$. We use proxy reward models, Beaver, Open Llama, and Tulu. As the gold reward model, we utilize Eurus.

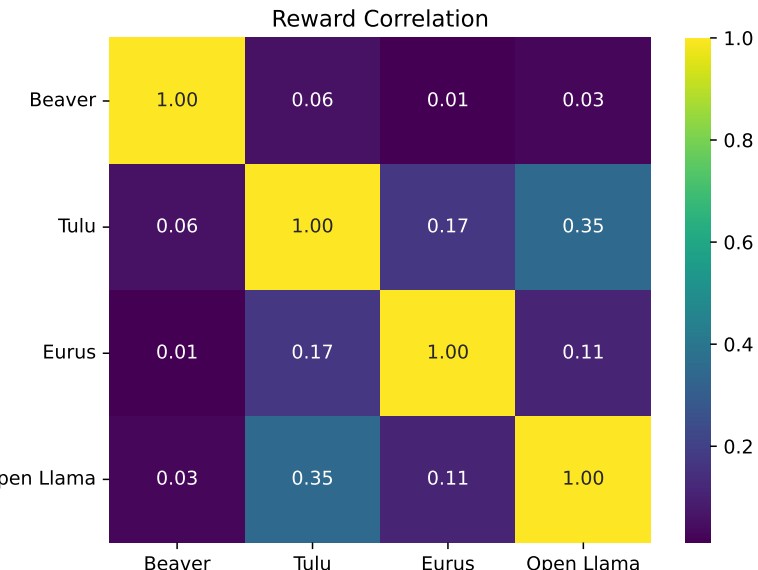

Figure 27: The average Spearman's rank correlation coefficient ($\rho$) between pairs of reward models in the Harmlessness dataset.

## G   Sentence Length Regularized BoN ($\mathrm{RBoN_L}$)

The objective function of $\mathrm{RBoN_L}$ (Sentence Length Regularized BoN) is given by:

$$y_{\mathbf{LBoN}}(x) = \arg\max_{y \in \mathcal{Y}_{\mathbf{ref}}} R(x, y) - \frac{\beta}{|y|}$$

where $\beta$ is a regularization parameter, $|y|$ denotes the token length of the sentence $y$ .

This approach aims to address the inherent bias toward shorter outputs often observed in a large language model we used in experiments. We now explain the rationale behind the specific form of the regularization term in $\mathrm{RBoN_L}$. Let $\mu$ represent a probability that is inversely proportional to the token length of the text $y$.

For example, we could define $\mu(y|x) = 1/|y|$ (e.g. $\mu(y'|x) = 1/|y'|, \mu(y''|x) = 1/|y''|...$), where $|y|$ represents the token length of output y.

**Definition G.1.** *We define a newly normalized distribution $\mu'$:*

$$\mu'(y \mid x) = \frac{1/|y|}{\sum_{\mathcal{Y}_{\mathbf{ref}}} \mu(\cdot \mid x)}$$

$$= \frac{1/|y|}{Z} \quad \left( \text{where } \sum_{\mathcal{Y}_{\mathbf{ref}}} \mu(\cdot \mid x) = Z \right)$$

**Proposition G.2.** *The objective function of $\mathrm{RBoN_L}$ is derived by considering the TV distance between the output probability $\mathbb{1}_y(\cdot \mid x)$ and $\mu'(\cdot \mid x)$ as a regularization term.*

*Proof.* Let us examine how the objective function of RBoN$_\text{L}$ is derived using definition G.1.

$$y_\text{LBoN}(x) = \underset{y \in \mathcal{Y}_\text{ref}}{\arg\max} \; R(x, y) + \beta \mathbf{TV} \left[ \mathbb{1}_y(\cdot \mid x) \| \mu'(\cdot \mid x) \right],$$

$$= \underset{y \in \mathcal{Y}_\text{ref}}{\arg\max} \; R(x, y) + \frac{\beta}{2} \sum_{y \in \mathcal{Y}_\text{ref}} |\mathbb{1}_y(\cdot \mid x) - \mu'(\cdot \mid x)|$$

$$= \underset{y \in \mathcal{Y}_\text{ref}}{\arg\max} \; R(x, y) + \frac{\beta}{2} \left( \left| 1 - \frac{1}{Z|y|} \right| + \underbrace{\frac{1}{Z|y'|} + \frac{1}{Z|y''|} + \cdots + \frac{1}{Z|y'''|}}_{=1 - \frac{1}{Z|y|}} \right)$$

$$= \underset{y \in \mathcal{Y}_\text{ref}}{\arg\max} \; R(x, y) + \beta \left( 1 - \frac{1}{Z|y|} \right)$$

$$= \underset{y \in \mathcal{Y}_\text{ref}}{\arg\max} \; R(x, y) - \frac{\beta}{Z|y|}$$

$$= \underset{y \in \mathcal{Y}_\text{ref}}{\arg\max} \; R(x, y) - \frac{\beta'}{|y|} \quad \left( \beta' = \frac{\beta}{Z} \right)$$

where $\beta'$ is a regularization parameter and **TV** denotes TV distance. $\square$

The purpose of this normalization is to counteract the effect of SRBoN$_\text{KL}$, which tends to favor shorter outputs. This formulation provides a theoretical basis for understanding how RBoN$_\text{L}$ achieves its length-aware behavior, and offers insight into its potential advantages over other decoding methods that may inadvertently bias toward shorter outputs.

Our methodological approach to assessing the divergence of output distributions from the length distribution $\mu'$ involves a comparative analysis of BoN sampling and RBoN$_\text{L}$. For each output y selected, we construct the corresponding $\mathbb{1}_y(\cdot \mid x)$ distribution. We then measure the TV distance between $\mathbb{1}_y(\cdot \mid x)$ and $\mu'(\cdot \mid x)$.

The results of this comparative analysis are visualized in Figure 28, Figure 29, and Figure 30.

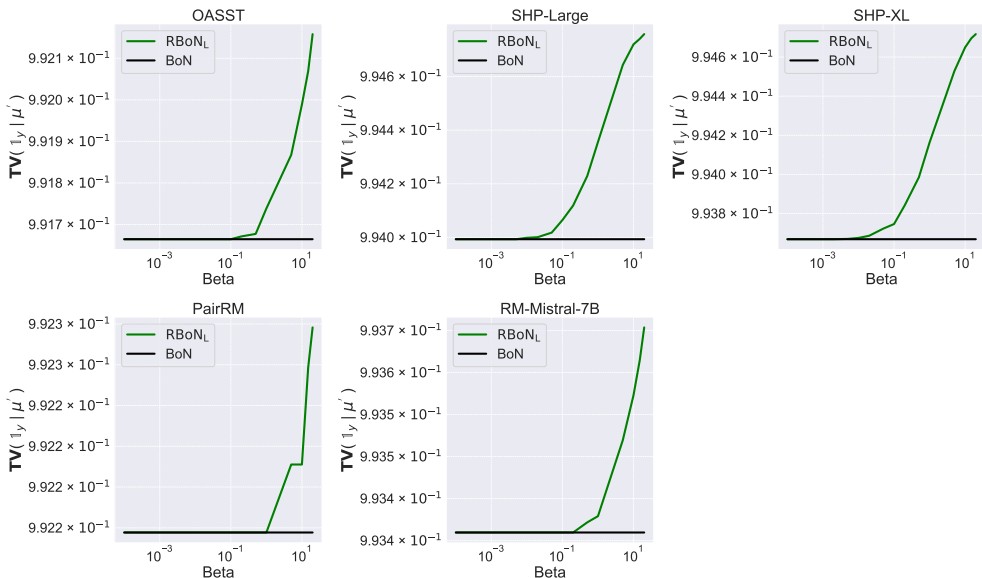

Figure 28: BoN sampling and RBoN$_L$ methods by measuring the TV distance between their output distributions and sentence length distribution $\mu'$ in AlpacaFarm. This allows us to evaluate how closely each method's outputs align with the desired distribution, with a smaller TV distance indicating a preference for shorter sentences.

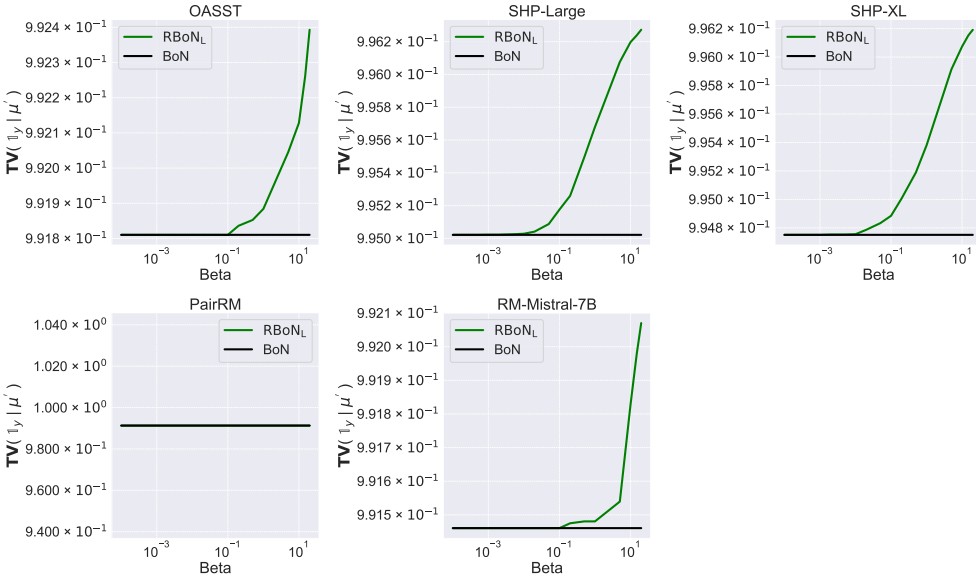

Figure 29: BoN sampling and RBoN$_L$ methods by measuring the TV distance between their output distributions and sentence length distribution $\mu'$ in Harmlessness. This allows us to evaluate how closely each method's outputs align with the desired distribution, with a smaller TV distance indicating a preference for shorter sentences.

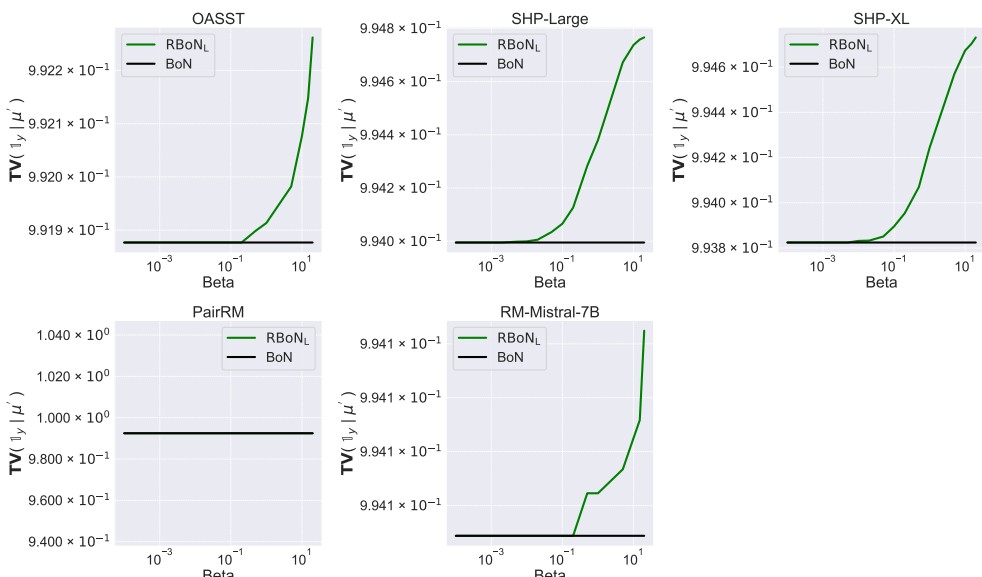

Figure 30: BoN sampling and RBoN$_\mathrm{L}$ methods by measuring the TV distance between their output distributions and sentence length distribution $\mu'$ in Helpfulness. This allows us to evaluate how closely each method's outputs align with the desired distribution, with a smaller TV distance indicating a preference for shorter sentences.

Our analysis shows that the output probability of RBoN$_\mathrm{L}$ deviates more from $\mu'$ than the output probability of BoN sampling. Table 7 illustrates the correlation between the length of the sequence and the values of gold reference reward (Eurus-RM-7B), focusing on subsets of sentences comprising the top 5, 10, 15 based on the proxy reward values. The strength of this correlation is an indication of the effectiveness of RBoN$_\mathrm{L}$; a stronger correlation indicates greater effectiveness of the method.

In Table 7, we have highlighted in **bold** the instances of high correlation compared to all samples used correlation, which corresponds to superior performance as shown in Table 2. In contrast, areas with lower correlation tend to show lower performance. This pattern shows a consistent relationship between correlation strength and method effectiveness. We also explored an alternative view of PairRM that had a high correlation but did not produce correspondingly strong results in Table 2.

We hypothesized that this discrepancy might be due to the range of the regularization parameter $\beta$. To investigate this hypothesis and to demonstrate the potential of RBoN$_\mathrm{L}$, we performed an extensive analysis by varying $\beta$ over a wide range, from 10 to 5000 Figure 31.

# H   Experiment with Qwen2.5-7B-Instruct

As an ablation study, we evaluate the methods using the Qwen (Qwen2.5-7B-Instruct) as the language model. Overall, we observe the same results as with Mistral-7B-SFT, where RBoN$_\mathrm{WD}$ outperforms the baseline algorithms (Figure 32).

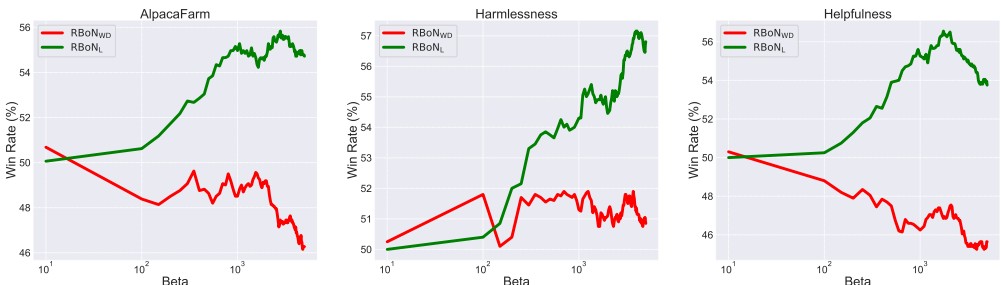

Figure 31: Performance analysis of RBoN$_L$ with varying $\beta$ (10 to 5000) across AlpacaFarm, Harmlessness, and Helpfulness datasets. PairRM and Eurus-RM-7B are used as proxy and gold reward models, respectively.

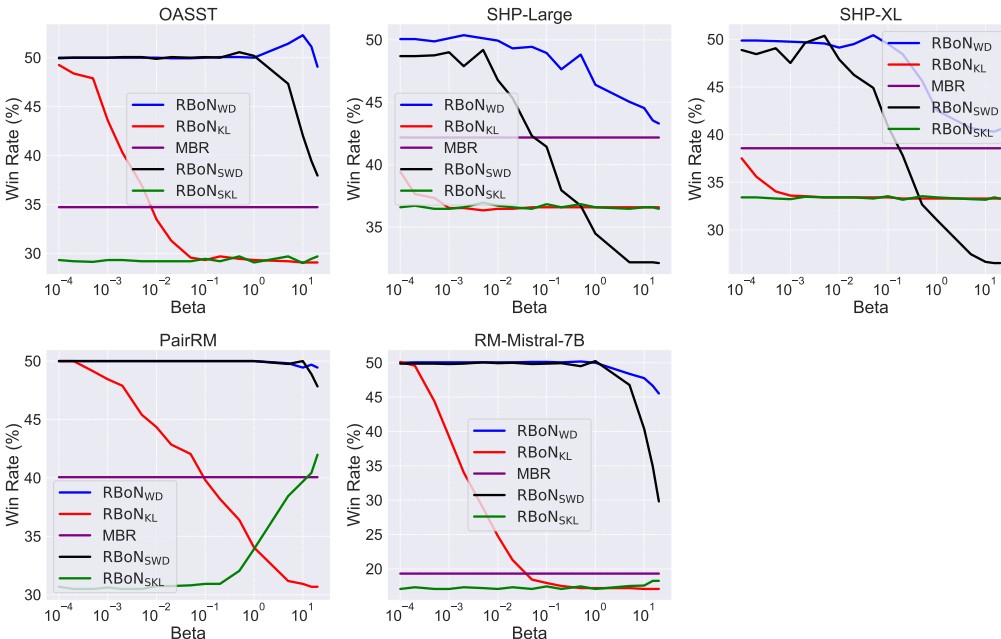

Figure 32: Evaluation of the RBoN method on the AlpacaFarm dataset with varying parameter $\beta$. We use proxy reward models, OASST, SHP-Large, SHP-XL, PairRM, and RM-Mistral-7B. As the gold reward model, we utilize Eurus-RM-7B, and Qwen as the language model.

Table 7: The correlation between sequence length and gold reference reward (Eurus-RM-7B) values, focusing on a subset of sentences that include the top 5, 10, 15 based on proxy reward values.

| Top N | OASST | SHP-Large | SHP-XL | PairRM | RM-Mistral-7B |
|---|---|---|---|---|---|
| **AlpacaFarm** | | | | | |
| **All** | 0.11 (0.33) | 0.11 (0.33) | 0.11 (0.33) | 0.11 (0.33) | 0.11 (0.33) |
| **5** | **0.27** (0.55) | -0.04 (0.56) | 0.05 (0.55) | 0.15 (0.59) | 0.10 (0.56) |
| **10** | **0.24** (0.44) | -0.02 (0.44) | 0.06 (0.41) | **0.17** (0.48) | 0.09 (0.56) |
| **20** | **0.21** (0.39) | -0.02 (0.37) | 0.06 (0.36) | **0.16** (0.41) | 0.08 (0.44) |
| **Harmlessness** | | | | | |
| **All** | 0.08 (0.45) | 0.08 (0.45) | 0.08 (0.45) | 0.08 (0.45) | 0.08 (0.45) |
| **5** | **0.24** (0.58) | 0.10 (0.57) | 0.13 (0.58) | **0.20** (0.62) | **0.37** (0.51) |
| **10** | **0.25** (0.50) | 0.11 (0.46) | 0.12 (0.47) | **0.19** (0.54) | **0.36** (0.41) |
| **20** | **0.22** (0.47) | 0.11 (0.41) | 0.11 (0.43) | **0.21** (0.49) | **0.34** (0.39) |
| **Helpfulness** | | | | | |
| **All** | 0.07 (0.40) | 0.07 (0.40) | 0.07 (0.40) | 0.07 (0.40) | 0.07 (0.40) |
| **5** | **0.28** (0.56) | -0.04 (0.58) | 0.11 (0.54) | **0.14** (0.62) | 0.06 (0.54) |
| **10** | **0.27** (0.47) | -0.05 (0.45) | 0.11 (0.42) | **0.15** (0.52) | 0.06 (0.40) |
| **20** | **0.24** (0.43) | -0.06 (0.40) | 0.10 (0.37) | **0.17** (0.46) | 0.03 (0.36) |

## I Reproducibility Statement

All datasets and models used in the experiments are publicly available (Table 8). Our code will be available as open source upon acceptance.

Table 8: List of datasets and models used in the experiments.

| Name | Reference |
| --- | --- |
| AlpacaFarm | Dubois et al. (2023) `https://huggingface.co/datasets/tatsu-lab/alpaca_farm` |
| Anthropic's hh-rlhf | Bai et al. (2022) `https://huggingface.co/datasets/Anthropic/hh-rlhf` |
| mistral-7b-sft-beta (Mistral) | Jiang et al. (2023a); Tunstall et al. (2023) `https://huggingface.co/HuggingFaceH4/mistral-7b-sft-beta` |
| Meta-Llama-3-8B-Instruct (Llama) | Dubey et al. (2024) `https://huggingface.co/meta-llama/Meta-Llama-3-8B-Instruct` |
| Qwen2.5-7B-Instruct (Qwen) | Yang et al. (2024); Team (2024) `https://huggingface.co/Qwen/Qwen2.5-7B-Instruct` |
| SHP-Large | Ethayarajh et al. (2022) `https://huggingface.co/stanfordnlp/SteamSHP-flan-t5-large` |
| SHP-XL | Ethayarajh et al. (2022) `https://huggingface.co/stanfordnlp/SteamSHP-flan-t5-xl` |
| OASST | Köpf et al. (2023) `https://huggingface.co/OpenAssistant/reward-model-deberta-v3-large-v2` |
| PairRM | Jiang et al. (2023b) `https://huggingface.co/llm-blender/PairRM` |
| RM-Mistral-7B | Dong et al. (2023) `https://huggingface.co/weqweasdas/RM-Mistral-7B` |
| Eurus-RM-7B | Yuan et al. (2024) `https://huggingface.co/openbmb/Eurus-RM-7b` |
| Beaver | Dai et al. (2024)`https://huggingface.co/PKU-Alignment/beaver-7b-v1.0-reward` |
| Tulu | Ivison et al. (2024) `https://huggingface.co/allenai/tulu-v2.5-ppo-13b-uf-mean-70b-uf-rm` |
| Open Llama | Diao et al. (2024) `https://huggingface.co/weqweasdas/hh_rlhf_rm_open_llama_3b` |
| MPNet | Song et al. (2020) `https://huggingface.co/sentence-transformers/all-mpnet-base-v2` |

