# OpenReview forum: "Evaluation of Best-of-N Sampling Strategies for Language Model Alignment"
_TMLR — Accepted by TMLR_

### Review · Reviewer_FU9k · 2024-10-31

**Summary Of Contributions:**

This paper improves the best-of-N samples for LLMs at test time, by introducing randomness into the previously proposed Regularized BoN algorithm. Specifically, it proves that the proposed stochastic methods are bounded theoretically, and leveraged two scalable open-source datasets to evaluate the method. It helps to improve LLM inference with minimum re-training efforts and potentially reduce bias on the reward model.

**Audience:**

Yes

**Broader Impact Concerns:**

I see no concerns regarding broader impact for this paper.

**Claims And Evidence:**

Yes

**Requested Changes:**

See weakness section above.

**Strengths And Weaknesses:**

**Strength:**

This paper is well organized. The English writing is proficient, all sections are explained logically in good clarity, and proofs of the proposed theorems are explained well.

The proposed methods show empirical improvements over previous RBoN methods, together with theoretical bound, which improved the validity of the proposed method.

**Weakness:**

Some claims could be stated in a more moderate way, e.g., "The RBoNL method is a significant contribution to the field of RBoN sampling" in conclusion, better to avoid "significant"

---

> ### Author Response · Authors · 2024-12-03
>
> We thank the reviewer for taking the time to read our manuscript carefully.
>
> ---
>
> ### Requested Changes
> >**Some claims could be stated in a more moderate way, e.g., "The RBoNL method is a significant contribution to the field of RBoN sampling" in conclusion, better to avoid "significant".**
>
> It was changed to appropriate text without 'significant'.
>
> Thank you for your valuable feedback.

---

### Review · Reviewer_uLRn · 2024-11-02

**Summary Of Contributions:**

This paper first systematically sutdy the existing BoN solutionss with regularization, and finds tht the regularization strategies in essence maximizes the worst case over a set of possible perturbation in the proxy reward. Inspired by this finding, this paper proposes stochastic RBoN, which has theoretically gurantee with the worst-case performance.

**Audience:**

Yes

**Claims And Evidence:**

Yes

**Requested Changes:**

1. Explicitly mention what is the regularization and what is the reference policy.

2. Discuss what is the reward model and how we can obtain it. Is it a reward model trained on the preference data like the reward model for RLHF?

I don't have a strong backgroud on BoN and reinforcement learning but the paper indeed seems to be a solid submission. All the requested change can be done with a minor revision.

**Strengths And Weaknesses:**

Strength:
1. The structure of this paper is good. The optimization is based on theoretical finding

2. The proposed solutions achieve SOTA solution.

3. The considered problem is new and can attract many audiences.


Weakness:

1. Some technical points are not clearly articulated.
     * Section 2.3..1, what is the reference policy here. Is it the original model output given x? How to calculate the KL distance between pi_y and pi_ref?
      * What does it mean by reward hacking? Does it mean that the reward model cannot accurately reflect the real value of the model's output? What cause this inaccuracy? Is that beacause of the data quality for training a reward model?

2. Some settings are not clear. For example, what is the reward model? Is it assumed to be known by this setting? How can we obtain such a reward model. I know this is not the main focus of the paper, but the authors should at least explicit it in an obvious position of the paper.

---

> ### Author Response · Authors · 2024-12-03
>
> We thank the reviewer for the valuable feedback.
>
> ---
>
> ###  Weakness
> > **Section 2.3..1, what is the reference policy here.**
>
> Thank you very much for pointing it out. The reference policy here is an instruction-tuned LLM that takes an input (i.e. instruction/prompt) x and returns an output y (i.e. response to the instruction/prompt). We will make this clear in the revision.
>
>
>  > **Is it the original model output given x?**
>
> Yes,  it is based on the original model output given x.
>
> > **How to calculate the KL distance between pi_y and pi_ref?**
>
> We compute KL-divergence between pi_y and pi_ref (the reference policy) as follows:
>
> $$
> D_{KL} [\pi_y (\cdot)|| \pi_{ref} (\cdot \mid x)]  = \sum_{y^\prime \in Y} \pi_y(y^\prime) \log \frac{\pi_y(y^\prime)}{\pi_{ref} (y^\prime \mid x)} = \sum_{y^\prime \in Y} 1[y=y^\prime\]  \log \frac{1}{\pi_{ref}(y \mid x)} = - \sum_{y^\prime \in Y} 1[y=y^\prime]  \log \pi_{ref}(y \mid x)
> $$
>
>
>
>
> > **What does it mean by reward hacking? Does it mean that the reward model cannot accurately reflect the real value of the model's output? What cause this inaccuracy? Is that beacause of the data quality for training a reward model?**
>
> Thank you very much for pointing it out.
>
> Reward hacking in reward models refers to the phenomenon where the model exploits weaknesses in the reward function to maximize rewards in unintended or inappropriate ways [1,2]. We will clarify it in the revision. Reward hacking occurs when the reward function does not fully capture the true objectives or desired outcomes.
>
> For example, Hadfield-Menell et al. (2017) [3] shows reward hacking that arises due to incorrect sensor readings at test time in a 10x10 navigation grid world.
>
> >**Some settings are not clear. For example, what is the reward model? Is it assumed to be known by this setting? How can we obtain such a reward model. I know this is not the main focus of the paper, but the authors should at least explicit it in an obvious position of the paper.**
>
> In this study, the reward model is used to measure the quality of text when making an assignment. We assume the availability of a proxy reward model, which approximates the true reward model but does not perfectly represent it.
>
> The reward model used is open access, as described in Appendix I.
>
> ---
>
> ### Requested Changes
>
> >**1. Explicitly mention what is the regularization and what is the reference policy.**
>
> We add "**The reference policy here takes an input x and returns a meaningful output y. The regularization described in this paper is like a penalty to stay away from the reference policy and the reference policy is the language model.**" in Section 2.3.1.
>
> >**2. Discuss what is the reward model and how we can obtain it. Is it a reward model trained on the preference data like the reward model for RLHF?**
>
> We add "**This reward model is used as a measure of the quality of the text when making an assignment. The reward model we consider in this paper is open access as described in Appendix H.**" in Section 2.2.
>
> We have made the corrections based on your feedback. Thank you for your valuable feedback.
>
> ---
> ### References
>
> [1] Pan, Alexander, Kush Bhatia, and Jacob Steinhardt. "The Effects of Reward Misspecification: Mapping and Mitigating Misaligned Models." *International Conference on Learning Representations*.
>
> [2] Lambert, Nathan, and Roberto Calandra. "The Alignment Ceiling: Objective Mismatch in Reinforcement Learning from Human Feedback." *arXiv preprint arXiv:2311.00168* (2023).
>
> [3] Hadfield-Menell, D., Milli, S., Abbeel, P., Russell, S. J., & Dragan, A. (2017). Inverse Reward Design. *Advances in Neural Information Processing Systems*, *30*.

---

### Review · Reviewer_GPCT · 2024-11-28

**Summary Of Contributions:**

This paper explores the optimization methods and limitations of "Best N Sampling" (BoN) used in the alignment process of LLMs. They proposed  an extension of RBoN framwork, SRBoN and the sentence Length Regularzied BoN. The paper analyzes how regularization techniques can alleviate the reward hacking problem. Through theoretical analysis and experimental evaluation, the study shows that SRBoN achieves a good performance.

**Audience:**

Yes

**Claims And Evidence:**

No

**Requested Changes:**

Add more experiments on different model series.

Add more baselines compared with DPO and PPO.

**Strengths And Weaknesses:**

Strengths:

(1) The paper is well-motivated and well-written.

(2) The paper's analysis is comprehensive and the results are strong, demonstrating a good performance across several benchmarks.

(3) It is great to see the proposed method has a good theoretical foundation.


Weaknesses:

(1) The model backbone used in the experiments is weak. Mistral 7B SFT is outdated and whether the proposed method could be applied to the latest model is a problem. I would suggest adding experiments on Llama-3.1 and Qwen-2.5. In addition, I am curious about whether the proposed method could be scaled up with larger models. It would be better to add the results with various model sizes.

(2) A minor question, there are many recent discussions on process reward models. I understand this paper mainly focuses on the outcome reward model and therefore only includes outcome reward models as baselines. I would appreciate to see discussion about the improvement compared with process reward models.

(3) Why not include DPO, PPO as baselines? It is too limited to compare with best-of-n methods only.

Typo: Section 4 Datasets - hh-rlhf datasets (HH) dataset, repeated "dataset".

---

> ### Author Response · Authors · 2024-12-04
>
> We thank the reviewer for the valuable feedback.
>
> We greatly appreciate the concerns raised about the experimental evaluation, as they are invaluable in helping us improve the experiments in our paper.
>
> ---
>
> ### Weaknesses
>
> > **The model backbone used in the experiments is weak. Mistral 7B SFT is outdated and whether the proposed method could be applied to the latest model is a problem. I would suggest adding experiments on Llama-3.1 and Qwen-2.5.**
>
> Thank you very much for pointing it out. We will add experiments using Llama and Qwen.
>
> We speculate that Regularized BoN strategies are also effective on stronger LLMs, as we find the performance of Mistral 7B SFT using RBoN_WD to be on par with the performance of Llama-3 using RBoN_WD (Figure 19, 20, and 21 in Appendix E).
>
>
> > **In addition, I am curious about whether the proposed method could be scaled up with larger models. It would be better to add the results with various model sizes.**
>
> Thank you very much for the suggestion. We will add experiments using Qwen-2.5 0.5B, 3B, and 7B to evaluate the scalability of the proposed method. We will revise the paper with the additional experiment once the computational resources are available.
>
> > **A minor question, there are many recent discussions on process reward models. I understand this paper mainly focuses on the outcome reward model and therefore only includes outcome reward models as baselines. I would appreciate to see discussion about the improvement compared with process reward models.**
>
> The theoretical guarantees presented in the paper assume that the reward is an outcome reward model, and the proof is not immediately transferable to process reward models. We will add a discussion of process reward models and clarify the limitations of our analysis.
>
>
> >**Why not include DPO, PPO as baselines? It is too limited to compare with best-of-n methods only.**
>
> Thank you very much for the suggestion. The purpose of the study is to evaluate the performance of inference-time alignment algorithms that do not require the training of the reference language model.
>
> That being said, we agree that it is valuable to evaluate the performance of inference-time alignment algorithms compared to learning-based alignment algorithms.
>
> We will conduct additional experiments and include the results of DPO, and PPO if computational resources are available.
>
> ---
>
> ### Requested Changes
>
> > **Add more experiments on different model series.**
>
> > **Add more baselines compared with DPO and PPO.**
>
> We will make the revisions based on your feedback as mentioned above. Thank you for your valuable feedback.

---

> > ### Author Response · Authors · 2024-12-11
> >
> > We have added the experiments with Qwen2.5-7B-Instruct on the AlpacaFarm dataset in Appendix H. We observe qualitatively the same results as with Mistral-7B-SFT, where RBoN_WD outperforms the baseline algorithms. We will include the results of Llama-3.2 once they become available.

---

### Decision · Action_Editor_WDzy · 2025-01-12

**Recommendation:** Accept with minor revision

**Comment:**

The reviewers are satisfied with the answers to their questions and the revisions made to the paper.  They unanimously recommend acceptance of the paper.  Recent work has emerged that should be cited. The authors are asked to add a discussion of the following works in the related work section.  The following two papers also optimize expected rewards subject to a KL divergence, but instead of reformulating the optimization as a min-max robustness optimization problem, they directly exploit the fact that there exists a closed form solution.  Hence, the optimization is simpler.

Mudgal, Sidharth, Jong Lee, Harish Ganapathy, YaGuang Li, Tao Wang, Yanping Huang, Zhifeng Chen et al. "Controlled Decoding from Language Models." In Forty-first International Conference on Machine Learning, 2024.

Khanov, M., Burapacheep, J., & Li, Y. ARGS: Alignment as Reward-Guided Search. In The Twelfth International Conference on Learning Representations, 2024.

The paper also lacks a limitation section.  The authors are asked to add a limitation section.  This section should at the very least discuss the cost of the optimization at inference time and acknowledge that there is a tradeoff in comparison to techniques that fine tune the LLM.  The proposed techniques do not fine tune the LLM, but inference time is increased while techniques that fine tune the LLM incur a one-time cost at training time while avoiding any overhead at inference time.

**Audience:**

This work will be of interest to researchers advancing alignment techniques for LLMs.

**Claims And Evidence:**

It was previously shown that LLM alignment techniques can suffer from overfitting to an inaccurate reward model.  This is especially problematic for techniques that rank generated text based on rewards without any regularization.  Previous work showed how to optimize KL and Wasserstein regularized expected rewards with approximate algorithms that return a deterministic policy.  In contrast, this paper shows that KL and Wasserstein regularized expected reward objectives have stochastic solutions.  It also derives algorithms to obtain approximately optimal stochastic policies.  Theoretical derivations support the claims and an empirical evaluation demonstrates the impact of regularization and stochasticity in practice.  This work advances the understanding of the community for regularized generation techniques.

---

> ### Author Response · Authors · 2025-02-07
>
> Dear Action Editor,
>
> Thank you for your positive feedback and for recommending our paper for acceptance with minor revisions. We also sincerely appreciate the thorough evaluation and constructive comments provided by all the reviewers, which have been invaluable in helping us improve our manuscript.
>
> We have carefully revised the manuscript in response to the concerns and suggestions, and the camera-ready version reflects these changes.